# The Effect of the Ethanolic Extracts from *Syzygium aromaticum* and *Syzygium nervosum* on Antiproliferative Activity and Apoptosis in HCT116 and HT-29 Cells

**DOI:** 10.3390/ijms26146826

**Published:** 2025-07-16

**Authors:** Thunyatorn Yimsoo, Weerakit Taychaworaditsakul, Sunee Chansakaow, Sumet Kongkiatpaiboon, Ngampuk Tayana, Teera Chewonarin, Parirat Khonsung, Seewaboon Sireeratawong

**Affiliations:** 1Department of Pharmacology, Faculty of Medicine, Chiang Mai University, Chiang Mai 50200, Thailand; thunyatorn_y@cmu.ac.th (T.Y.); parirat.khons@cmu.ac.th (P.K.); 2Laboratory Animal Center, Thammasat University, Rangsit Campus, Pathum Thani 12121, Thailand; 3Department of Biochemistry, Faculty of Medicine, Chiang Mai University, Chiang Mai 50200, Thailand; weerakit.tay@cmu.ac.th (W.T.); teera.c@cmu.ac.th (T.C.); 4Faculty of Pharmacy, Chiang Mai University, Chiang Mai 50200, Thailand; sunee.c@cmu.ac.th; 5Drug Discovery and Development Center, Thammasat University, Pathum Thani 12121, Thailand; s_u_m_e_t@hotmail.com (S.K.); ngampuk@tu.ac.th (N.T.); 6Thammasat University Research Unit in Cannabis and Herbal Product Innovation, Thammasat University, Rangsit Campus, Pathum Thani 12121, Thailand; 7Clinical Research Center for Food and Herbal Product Trials and Development (CR-FAH), Faculty of Medicine, Chiang Mai University, Chiang Mai 50200, Thailand

**Keywords:** *Syzygium aromaticum*, *Syzygium nervosum*, p53-wild-type CRC, antiproliferative, apoptosis

## Abstract

Colorectal cancer (CRC) is the third most diagnosed cancer worldwide, and p53 dysfunction plays a significant role in its pathogenesis by impairing cell cycle control and apoptosis. This study aimed to elucidate the phytochemical composition and anticancer potential of extract of residue from clove hydrodistillation (*Syzygium aromaticum*, SA) and seed extract from *Syzygium nervosum* (SN). LC-DAD-MS/MS analysis identified gallic acid (2.68%) and ellagic acid (6.70%) as major constituents in SA, while SN contained gallic acid (0.26%), ellagic acid (3.06%), and 2′,4′-dihydroxy-6′-methoxy-3′,5′-dimethylchalcone (DMC) as major constituents. Both extracts exhibited potent antioxidant effects as evidenced by DPPH and ABTS assays. In vitro assays showed that SA and SN significantly inhibited the proliferation of HCT116 (p53 wild-type) colorectal cancer cells, with minimal effects on HT-29 (p53 mutant) cells. Apoptosis was confirmed in HCT116 via Annexin V-FITC/PI staining and increased caspase-3/7 activity. Cell cycle analysis revealed sub-G1 accumulation, accompanied by upregulated p21 and concurrently downregulated cyclin D1 expression, both hallmarks of p53-mediated checkpoint activation. These molecular effects were not observed in HT-29 cells. In conclusion, SA and SN extracts selectively induce apoptosis and cell cycle arrest in p53-functional CRC cells, likely mediated by their phenolic constituents. These findings support their potential as promising plant-derived therapeutic agents for targeted colorectal cancer treatment.

## 1. Introduction

Colorectal cancer (CRC) remains a significant public health issue worldwide, with its incidence steadily rising in numerous regions. It is one of the five leading malignancies affecting both men and women globally [1]. Due to its aggressive nature and late-stage diagnosis, CRC remains the main cause of cancer-related mortality in Thailand (WHO, 2023), where the incidence of CRC has been steadily increasing. Currently, surgical resection is the primary treatment for early-stage CRC, but only a small portion of patients are entitled, meaning there is a slow generation of evidence on its effect [2]. Most advanced-stage CRC cases require chemotherapy, which often face challenges due to tumor resistance mechanisms [3]. Unfortunately, CRCs can develop resistance to conventional chemotherapy through genetic mutations and adaptive cell responses, limiting the effectiveness of treatment [4]. Furthermore, chemotherapy is often associated with many side effects that can have a significant impact on the quality of life of a patient. Therefore, the development of new targeted therapies with improved efficacy and reduced side effects is essential for improving the outcomes of colon cancer [5].

Apoptosis and antiproliferation are important mechanisms in cancer treatment, acting synergistically to suppress tumor growth. Apoptosis, a regulated mechanism of programmed cell death, is essential for preventing tissue and tumor development [6,7]. It occurs through extrinsic and intrinsic pathways, with caspases playing a central role in cell degradation. In cancer treatment, inducing apoptosis can help eliminate tumor cells and helps overcome resistance to chemotherapy [8]. Antiproliferation strategies, on the other hand, focus on inhibiting uncontrolled cell division by targeting key regulators of the cell cycle, such as cyclins and CDKs [9]. Combining apoptotic induction with antiproliferative strategies enhances treatment efficacy by both eliminating cancer cells and preventing their further proliferation [10]. Understanding the interplay between these mechanisms is crucial for developing effective cancer therapies.

Plants in the *Syzygium* genus contain a wide range of polyphenolic compounds that have shown promising anticancer properties. Among its many species, *Syzygium aromaticum* (L.) Merr. & L.M.Perry, commonly referred to as clove, and *Syzygium nervosum* DC (syn. *Cleistocalyx nervosus* Kosterm., and *Eugenia nervosa* Bedd., locally known in Thai as Makiang) are prominent in Southeast Asia, including Thailand. The primary bioactive compound in *S. aromaticum* is eugenol, which consists of approximately 50% of clove essential oil and has been reported to suppress tumor progression through the induction of G1-phase arrest in colon cancer cells [11]. Interestingly, the residue remaining after the hydrodistillation of *S. aromaticum,* typically discarded as waste, was found in this study to retain high levels of phenolic compounds, particularly gallic acid and ellagic acid. Both compounds exhibit well-established antioxidant and anticancer activities [12]. Despite this, the therapeutic potential of this clove residue extract (hereafter referred to as SA extract) against colorectal cancer has not been thoroughly investigated. Similarly, *S. nervosum*, a native plant species found in Thailand and surrounding regions, is rich in diverse phytochemicals, including anthocyanins, hydrolyzable tannins, caffeoylquinic acid, gallic acid, and ellagic acid [13]. Notably, 2′,4′-Dihydroxy-6′-methoxy-3′,5′-dimethylchalcone (DMC), a chalcone derivative isolated from the seeds of *Syzygium nervosum*, has been shown to exhibit chemopreventive activity in animal models of colorectal cancer [14].

In this study, we investigated the phytochemical profiles and in vitro anticancer activities of two plant-derived extracts: the clove residue from the hydrodistillation of *S. aromaticum* (SA extract) and the seed extract of *S. nervosum* (SN extract). Two colorectal cancer cell lines with differing p53 statuses were employed: HCT116 (wild-type p53) and HT-29 (mutant p53). We evaluated the antiproliferative effects of these extracts and investigated the underlying mechanisms involved in apoptosis induction. The findings may support the development of phytochemical-based therapeutic candidates for CRC treatment.

## 2. Results

### 2.1. Quality Control of the Raw Materials

The physicochemical properties of the raw material were evaluated according to the methods outlined in the Thai Herbal Pharmacopoeia (2021) [15]. The results met the established criteria. The raw materials that had been evaluated were then extracted for further processing.

### 2.2. Quality Control of the Extracts

#### 2.2.1. Identification and Quantification of Major Components

Both qualitative and quantitative analyses of the samples tested were conducted in this study. The phytochemical profiles of *S. aromaticum* and *S. nervosum* were analyzed using a liquid chromatography coupled with a diode array detector and tandem mass spectrometry (LC-DAD-MS/MS) method. The representative chromatogram of *S. aromaticum* is shown in Figure 1A–C, while *S. nervosum* is shown in Figure 1D–F. Mass spectral data are illustrated in Table 1 and Table 2. Peak identification was performed by comparing mass spectral data with the NIST and mzCloud databases, also shown in Table 1 and Table 2.

The predominant constituents identified in the *S. aromaticum* extract were gallic acid (SA1), chlorogenic acids (SA4–SA5), and ellagic acid (SA7) while those in *S. nervosum* were gallic acid (SN1), pedunculagins (SN2–SN3), ellagic acid (SN4), and 2′,4′-dihydroxy-6′-methoxy-3′,5′-dimethylchalcone (SN7). In addition, there were number of unidentified compounds. Gallic acid (SA1 and SN1) could explicitly be identified from [M−H]^−^ ions at *m*/*z* = 169.0140 (calculated 169.01315) and [M+H]^+^ ions at *m*/*z* = 171.0288 (calculated 171.02880). Fragments of gallic acid at *m*/*z* of 125 from the precursor of *m*/*z* 169 in the negative mode and *m*/*z* of 127 from the precursor of *m*/*z* 171 in the positive mode were related to the CO_2_ (44 Da) neutral loss due to the α mechanism, as described in the literature [16]. Ellagic acid (SA7 and SN4) could explicitly be elucidated from the [M−H]^−^ ion at an *m*/*z* of 300.9987 (calculated 300.99789), [M+H]^+^ ion at *m*/*z* of 303.0134 (calculated 303.01354), and [M+Na]^+^ ion at *m*/*z* of 324.9954 (calculated 344.99549). Fragments of ellagic acid at *m*/*z* of 257 from the precursor of *m*/*z* 301 in the negative mode were also related to the CO_2_ (44 Da) neutral loss. Identification of gallic acid and ellagic acid was also confirmed using co-chromatography with their authentic standards. Biflorin and isobiflorin (SA4 and SA5) could tentatively be identified from [M−H]^−^ ions at an *m*/*z* of 353.0870 (calculated 353.08671) and [M+H]^+^ ions at an *m*/*z* of 355.1020 (calculated 355.10236). These compounds have been isolated from *S. aromaticum* by Cai and Wu (1996) [17] and Tanaka et al. (1993) [18]. Strictinin, an ellagitannin, could be identified from the [M−H]^−^ ion at an *m*/*z* of 633.0729 (calculated 633.07224) and the [M+Na]^+^ ion at an *m*/*z* 657.0695 (calculated 657.06983). Fragmentation with a precursor of *m*/*z* 633.0729 strictinin in the negative mode showed the signal of ellagic acid at *m*/*z* 300.9988. Pedunculagins and their isomers (SN2 and SN3), a class of ellagitannins widely found in plants [19], could be detected in [M−2H]^2−^ ions at *m*/*z* 391.0303 (calculated 391.03068), [M−H]^−^ ion at *m*/*z* 783.0677 (calculated 783.06755), [M+H]^+^ ion at *m*/*z* 785.0827 (calculated 785.08320), and [M+NH_4_]^+^ ions at *m*/*z* 802.1097 (calculated 802.01975). Pedunculagins’ fragment with *m*/*z* 300.9984 from precursors of 391.0303 and 783.0677, and *m*/*z* 303.0134 from precursors of 799.0986 and 802.1094, were related to the ellagic acid fragments. 2′,4′-Dihydroxy-6′-methoxy-3′,5′-dimethylchalcone (DMC, SN7) could tentatively be identified by [M−H]^−^ ions at *m*/*z* 297.1127 (calculated 297.11323), [M+H]^+^ ions at *m*/*z* 299.12779, and [M+Na]^+^ ions at *m*/*z* 321.1094 (calculated 321.10973). This compound has been isolated from *S. nervosum* fruits and seeds [20,21]. DMC’s dimers (SN5 and SN6) could tentatively be identified from [M−2H]^2−^ ions at *m*/*z* 297.1130 (calculated 297.11323), [M+2H]^2+^ ions at *m*/*z* 299.1275 (calculated 299.12779), and [M+Na]^+^ ions at *m*/*z* 619.2299 (calculated 619.23024). Fragment of dimers showed the presence of the DMC signal at *m*/*z* 297 in the negative mode.

For quantitative evaluation, gallic acid and ellagic acid served as chemical markers in the analysis of the tested extracts. The quantification of gallic acid and ellagic acid was performed using the data obtained from a diode array detector (DAD). Calibration curves were generated by plotting peak areas against standard concentrations using the least squares method. Gallic acid showed a good linear relationship in the range of 15.625–1000 μg/mL with a coefficient of determination (r^2^) of 0.9999 (Equation: Y = 0.1656X + 0.3767). Ellagic acid showed an acceptable linear relationship in the range of 15.625–1000 μg/mL with a coefficient of determination (r^2^) of 0.9958 (Equation: Y = 0.3531X−10.242). Quantitative analysis showed that the *S. aromaticum* extract contained gallic acid and ellagic acid at concentrations of 2.68% and 6.70% *w*/*w*, respectively, while the corresponding levels in the *S. nervosum* extract were 0.26% and 3.06% *w*/*w*.

#### 2.2.2. Antioxidant Properties of SA and SN Extracts

The antioxidant capacity of *S. aromaticum* (SA) and *S. nervosum* (SN) extracts was assessed using DPPH and ABTS radical scavenging assays. SA demonstrated the strongest activity, with IC_50_ values of 8.75 ± 0.37 µg/mL (DPPH) and 10.23 ± 0.75 µg/mL (ABTS), followed by SN with values of 12.92 ± 0.30 and 11.93 ± 0.25 µg/mL, respectively. Importantly, both extracts exhibited a greater scavenging capacity than vitamin C, which showed IC_50_ values of 16.09 ± 3.57 and 13.20 ± 0.01 µg/mL for DPPH and ABTS, respectively (Figure 2). This enhanced antioxidant activity corresponded with the higher total phenolic content observed in SA (308.14 ± 3.52 mg GAE/g extract) compared to SN (236.08 ± 4.28 mg GAE/g extract). These results suggest that the polyphenol-rich composition of both extracts contributes to their antioxidant potential, which may play a role in modulating oxidative stress in colorectal cancer cells. To further investigate their biological relevance, we next examined the cytotoxic effects of SA and SN extracts in both colorectal cancer and normal cell lines.

### 2.3. Cytotoxic Effects of SA and SN Extracts on Colorectal Cancer and Normal Cells

The cytotoxic potential of SA and SN extracts was evaluated in HCT116 (p53 wild-type) and HT-29 (p53 mutant) colorectal cancer cell lines using the MTT assay. As shown in Figure 3A–D, both extracts significantly reduced cell viability in a dose- and time-dependent manner. In HCT116 cells, SA induced a marked reduction in viability beginning at 100 µg/mL, with the most pronounced effects observed at 72 h (Figure 3A). SN exhibited a similar trend (Figure 3B), although higher concentrations were generally required to achieve comparable cytotoxicity. In HT-29 cells, both extracts also reduced viability, with significant effects evident at 48 and 72 h, particularly at concentrations ≥ 200 µg/mL (Figure 3C,D). These results were corroborated by IC_50_ values (Table 3, Figure 4), which consistently demonstrated a greater potency of SA over SN across all time points. To assess selectivity, cytotoxicity was further evaluated in normal lung fibroblast MRC-5 cells (Figure 3E,F), which are commonly used as a non-cancerous controls in cytotoxicity testing. IC_50_ values were obtained only at 72 h, enabling calculation of the Selectivity Index (SI), defined as the ratio of IC_50_ in MRC-5 to that in cancer cells. SA showed higher selectivity in HCT116 cells (SI = 2.24) compared to HT-29 (SI = 1.23), while SN showed SI values of 2.59 and 1.84, respectively. To compare the selectivity of both extracts, a bar graph summarizing IC_50_ values ± SD at 72 h was generated across HCT116, HT-29, and MRC-5 cells (Figure 4A). The data show that both SA and SN were more cytotoxic toward colorectal cancer cells than normal fibroblasts, particularly in HCT116 cells. Although the IC_50_ difference between HT-29 and MRC-5 was less pronounced, HT-29 remained more sensitive than MRC-5, especially for SN (HT-29: 232.92 ± 8.08 µg/mL; MRC-5: 428.3 ± 110.6 µg/mL). Collectively, these findings confirm that both extracts exert dose- and time-dependent cytotoxic effects, with SA demonstrating superior efficacy and selectivity. The preferential cytotoxicity toward both wild-type and mutant p53 CRC cells, particularly the p53-wild-type HCT116, and the relatively lower toxicity in normal cells support the potential development of SA and SN as selective anticancer agents.

### 2.4. SA and SN Extracts Induce Apoptosis in Colorectal Cancer Cells

#### 2.4.1. Flow Cytometer for Apoptosis Assay Annexin V-FITC/PI

To investigate whether the cytotoxic effects of SA and SN extracts were associated with apoptosis, Annexin V-FITC/PI double staining followed by flow cytometric analysis was performed in HCT116 and HT-29 colorectal cancer cell lines. As shown in Figure 5A,B, treatment with both SA and SN extracts led to a dose-dependent increase in early apoptotic cell populations in HCT116 cells, with SA demonstrating a more potent pro-apoptotic effect. Especially, SA at 100 and 200 µg/mL significantly elevated early apoptosis to over 40%, compared to less than 10% in untreated controls. SN extract also induced a marked increase in early apoptosis, although the extent was slightly lower than that observed with SA. In contrast, late apoptotic and necrotic populations remained minimal across all treatment conditions, indicating that early-stage apoptosis was the predominant mode of cell death. In HT-29 cells (Figure 5C,D), both extracts similarly induced early apoptosis in a concentration-dependent manner, but the overall response was substantially lower than that seen in HCT116 cells. SA extract, particularly at 100 and 200 µg/mL, showed a significant increase in early apoptotic cells, whereas SN induced a relatively modest apoptotic response across all concentrations tested. These findings suggest that SA and SN extracts exert their cytotoxic effects, at least in part, through the induction of apoptosis, primarily at early stages. Importantly, HCT116 cells (p53 wild-type) were more susceptible to apoptosis induction than HT-29 cells (p53 mutant), highlighting potential differences in p53-dependent apoptotic pathways. The quantified apoptosis data are summarized in Appendix A.

#### 2.4.2. Caspase 3/7 In Situ Fluorescence Microscope

To further evaluate the pro-apoptotic effects of SA and SN extracts, caspase-3/7 activity was visualized using a FLICA-based in situ fluorescence assay. Under fluorescence microscopy, cells treated with SA and SN extracts at 200 µg/mL showed more prominent green fluorescence signals compared to untreated controls. These signals, corresponding to active caspase-3/7, were primarily localized in the cytoplasm of affected cells and appeared alongside blue Hoechst 33342-stained nuclei. Quantitative analysis revealed a dose-dependent increase in fluorescence intensity. In HCT116 cells, SA extract resulted in the highest signal at 200 µg/mL (41,660 ± 4256 AU), while SN treatment at the same concentration produced a lower but significant increase (7431 ± 611 AU) when compared with the control shown in Figure 6A. In HT-29 cells, activation of caspase-3/7 was also measured, with maximum intensities of 25,780 ± 4745 AU for SA and 18,939 ± 1135 AU for SN. All increases were statistically significant when compared with the control, as summarized in Figure 6B and Appendix A. Overall, the results indicate that both SA and SN extracts effectively induce apoptosis through caspase-3/7 activation in colorectal cancer cells, with a stronger effect detected in the HCT116 cells (p53 wild-type) than in the HT-29 cells (p53 mutant).

### 2.5. Inhibition of Cell Growth and Induction of Cell Cycle Arrest by SA and SN Extracts in Colorectal Cancer Cells

#### 2.5.1. Colony Formation Assay

To assess the long-term antiproliferative potential of SA and SN extracts, a colony formation assay was performed in HCT116 cells and HT-29 cells. Cells were exposed to an increased extract concentration (25–200 µg/mL), followed by incubation to allow colony development. As shown in Figure 7A, SA extract significantly diminished the colony-forming efficiency of HCT116 cells in a concentration-dependent manner, with the highest inhibition observed at 200 µg/mL (10.80 ± 5.84%). Similarly, Figure 7C demonstrates that SA also suppressed colony formation in HT-29 cells, although to a lesser extent. In contrast, SN extract showed only moderate inhibition in HCT116 cells (Figure 7B), with a significant reduction at 200 µg/mL (72.58 ± 3.43%), and had minimal effects on HT-29 cells (Figure 7D), where colony formation remained largely unchanged across all concentrations. Quantitative analysis of colony-forming efficiency is presented in Table 4, confirming that HCT116 cells exhibited greater sensitivity to both extracts compared to HT-29 cells. These findings indicate that the SA extract exhibits stronger and broader inhibitory effects on colorectal cancer cell clonogenicity, particularly in the p53 wild-type HCT116 line, while SN extract has limited efficacy in HT-29 cells.

#### 2.5.2. Cell Cycle Analysis by Flow Cytometry and Western Blotting

To determine whether the antiproliferative effects of SA and SN extracts were associated with cell cycle disruption, flow cytometric analysis was conducted on HCT116 and HT-29 cells following 24 h treatment. As shown in Figure 8A,B, treatment with SA and SN extracts led to a dose-dependent increase in the sub-G1 population in HCT116 cells, indicative of DNA fragmented cells. The increase in sub-G1 fraction was most prominent at concentrations ≥ 100 µg/mL. In HT-29 cells, a significant rise in sub-G1 population was detected only at the highest concentration (200 µg/mL) of either extract (Figure 8C,D), indicating lower sensitivity compared to HCT116 cells. To explore the molecular basis of sub-G1 arrest, the expression levels of the cell cycle regulators p21 and cyclin D1 were examined by Western blot analysis (Figure 9). In HCT116 cells, SA extract markedly upregulated p21 with a dose-dependent decrease in cyclin D1 expression. SN extract significantly suppressed cyclin D1 levels but had minimal effect on p21 expression. These molecular alterations correspond to the observed sub-G1 accumulation and reduced colony formation, indicating that both extracts may induce G1/S checkpoint activation leading to apoptosis. In contrast, HT-29 cells showed the opposite expression pattern following SA treatment, with reduced p21 and elevated cyclin D1 levels, particularly at higher concentrations. SN extract had little effect on either protein. These findings are consistent with the aggressive, apoptosis-resistant nature of HT-29 cells and reinforce the observed selective efficacy of SA and SN extracts in p53 wild-type HCT116 cells.

## 3. Discussion

This study aimed to explore the phytochemical constituents, antioxidant capacities, and anticancer activities of ethanol extracts obtained from the hydrodistillation residue of *S. aromaticum* (SA) and the seeds of *S. nervosum* (SN), using two colorectal cancer cell lines with different p53 statuses—wild-type HCT116 and mutant HT-29.

The LC-DAD-MS/MS analysis revealed that both extracts were rich in polyphenolic compounds, particularly gallic acid and ellagic acid, which were confirmed through co-chromatography following accepted standards. These polyphenols are widely recognized for their strong antioxidant, anti-inflammatory, and anticancer activities, offering a mechanistic rationale for the biological effects observed in this study [22,23]. Gallic acid, a well-known phenolic compound present in both SA and SN extracts, has been reported to inhibit colorectal cancer cell proliferation and induce apoptosis through activation of intrinsic apoptotic pathways [24]. It also exhibits similar effects in non-small-cell lung carcinoma cells by promoting apoptosis and modulating cell cycle regulatory proteins [25]. The quantification showed that SA extract contained a higher concentration of gallic acid (2.68%) than SN (0.26%). Similarly, ellagic acid was detected at a higher concentration in the SA extract (6.70%) compared to the SN extract (3.06%). This compound has been demonstrated to induce G0/G1-phase arrest and to trigger apoptotic cell death in HCT116 colorectal cancer cells [26]. Consistently, ellagic acid has been shown to modulate the expression of genes related to proliferation, apoptosis, and angiogenesis in HCT116 colorectal cancer cells [27]. In addition to these major compounds, the SA extract also contained chlorogenic acids and biflorin derivatives (SA4–SA5), which have been reported to exert cytotoxic effects on colorectal cancer cells [28]. In contrast, the SN extract was characterized by the presence of pedunculagins (SN2–SN3), a subclass of ellagitannins [19], and 2′,4′-dihydroxy-6′-methoxy-3′,5′-dimethylchalcone (DMC, SN7), which are known as polyphenolic compounds commonly found in plants rich in polyphenols [21]. Importantly, both DMC (SN7) and its dimers (SN5–SN6), which were also identified in *S. nervosum* seeds in the previous studies [14,29,30], have been shown to exert pro-apoptotic effects by suppressing Bcl-2 and activating caspase-3 in cancer cells [30,31]. These findings suggest that gallic acid and ellagic acid of both SA and SN extracts serve as key bioactive markers that may contribute to anticancer activity.

Based on these findings, the total phenolic content (TPC) of each extract was assessed in parallel with antioxidant activity, given that phenolic-rich plant extracts are known to exhibit strong antioxidant properties attributed to their redox potential [32,33]. TPC was assessed via the Folin–Ciocalteu assay, a widely accepted method for estimating polyphenol levels in plant-derived samples [34,35]. In this study, SA extract exhibited a higher phenolic content (308.14 ± 3.52 mg gallic acid/g extract) compared to SN extract (236.08 ± 4.28 mg gallic acid/g extract). This observation supports previous reports indicating that clove is particularly rich in phenolic compounds [36]. Moreover, TPC values exceeding 100 mg GAE/g are commonly associated with strong antioxidant potential [37,38]. Thus, DPPH and ABTS radical scavenging assays were performed to assess the antioxidant activity of both extracts and to confirm their free radical-neutralizing potential [39,40]. SA and SN extracts demonstrated IC_50_ values of 8.75 ± 0.37 and 12.92 ± 0.30 µg/mL, respectively, in the DPPH assay, while values of 10.23 ± 0.37 and 11.93 ± 0.25 µg/mL were observed in the ABTS assay (Figure 2). All values were well below 50 µg/mL, indicating strong and effective radical scavenging activity [41,42]. Therefore, SA exhibited a greater radical scavenging potential than SN, which is likely attributable to its higher phenolic content. The previous studies have suggested that antioxidant activity may contribute to its enhanced anticancer potential [43,44,45].

The present findings demonstrate that both SA and SN extracts exert dose- and time-dependent cytotoxic effects against colorectal cancer cells. Interestingly, HCT116 (p53 wild-type) cells showed greater responsiveness to both extracts than HT-29 (p53 mutant) cells, suggesting a role for the p53 status in modulating sensitivity. SA consistently exhibited higher antiproliferative potency than SN across all time points. Importantly, both extracts displayed preferential cytotoxicity toward cancer cells over normal lung fibroblasts (MRC-5), as demonstrated using the Selectivity Index (SI), which is defined as the ratio of IC_50_ in normal to cancer cells. An SI value greater than two is generally considered acceptable for selective anticancer agents [46,47]. In our study, both SA (SI = 2.24) and SN (SI = 2.59) exceeded this threshold in HCT116 cells. Although IC_50_ values in MRC-5 could only be determined at 72 h, the higher values relative to cancer cells indicate reduced susceptibility. These findings support selective toxicity and are consistent with prior reports on plant-derived compounds showing minimal effects on non-malignant cells [21,48]. To enhance translational relevance, we further evaluated the response of HT-29 cells, representing a chemoresistant, p53-mutant CRC phenotype, in comparison with MRC-5. While the selectivity was less prominent than in HCT116, SN still showed moderate selectivity (SI = 1.84), and SA retained a lower but notable SI of 1.23, indicating partial discrimination between malignant and normal cells even in the presence of p53 mutations. Interestingly, when the extracts were applied at the IC_50_ concentration determined for the more resistant HT-29 cells, they not only effectively reduced the viability of HT-29 but also significantly affected the more sensitive HCT116 cells, which express wild-type p53. Notably, this concentration did not exhibit cytotoxic effects on MRC-5 fibroblasts, further supporting the safety margin of the compounds. This observation highlights the practical advantage of selecting a working concentration based on resistant cancer cells, enabling broader therapeutic efficacy while minimizing off-target toxicity. Such an approach may enhance the effectiveness of treatment strategies and guide rational dose selection in future preclinical studies. The inclusion of a direct IC_50_ and SI comparison across all cell lines (Figure 4) reinforces the selective anticancer potential of these polyphenol-rich extracts. Their ability to target cancer cells, including those with p53 mutations, while sparing normal cells, underscores their promise as candidates for further development. Further investigations are warranted to clarify the molecular mechanisms involved, particularly p53-dependent signaling, as well as apoptosis- and cell cycle-related pathways underlying the cytotoxic effects observed in colorectal cancer cells.

To evaluate the anticancer potential of SA and SN extracts, we assessed their effects on two colorectal cancer cell lines, HCT116 and HT-29 cells, which differ in p53 status. HCT116 cells express wild-type p53, a key mediator of cell cycle arrest and apoptotic response to DNA damage, while HT-29 cells carry a mutant form of p53, resulting in impaired tumor-suppressive functions [49]. Therefore, to examine apoptosis induction, two methods were performed: flow cytometric analysis with Annexin V-FITC/PI staining and in situ fluorescence microscopy for caspase-3/7 activity. Initially, flow cytometric analysis was conducted using Annexin V-FITC/PI staining, a well-established and reliable technique for distinguishing viable cells from those undergoing early apoptosis, as well as late apoptosis or necrosis [50]. Consistent with their phenolic profiles, both SA and SN extracts induced apoptosis in a dose-dependent manner, with a more prominent response in HCT116 than HT-29 cells. The SA concentration at 100 and 200 µg/mL increased the proportion of early apoptotic HCT116 cells to over 40%, while SN produced a comparable but slightly weaker effect. The minimal presence of late apoptotic and necrotic cells indicates that early-stage apoptosis was the dominant form of cell death. These results align with earlier studies indicating that the bioactive components of *S. aromaticum* suppress cell proliferation and trigger apoptosis in HCT116 cells via modulation of the PI3K/Akt/mTOR signaling cascade [51]. Similarly, other studies have shown that compounds isolated from *S. nervosum*, particularly DMC, induce apoptosis in colorectal cancer cells by downregulating Bcl-2 and activating caspase-3-dependent apoptotic signaling [14]. In HT-29 cells, apoptosis induction was observed in both extracts but remained less pronounced, emphasizing the role of p53. Notably, SA consistently showed greater efficacy than SN inducing apoptosis at equivalent doses, highlighting a potential link between SA’s activity and p53 functionality.

To confirm apoptosis induction, caspase-3/7 in situ fluorescence microscopy was assessed. This technique directly visualizes apoptosis at the single-cell scale and supports the Annexin V staining results. The assay was carried out using a fluorescently labeled caspase-3/7 inhibitor that irreversibly binds to active enzymes in apoptotic cells, enabling their detection under a fluorescence microscope. The resulting green fluorescence provides both spatial and morphological information about cells undergoing apoptosis [52,53]. Green fluorescence signals representing caspase activity were stronger in SA-treated HCT116 cells compared to SN. This observation aligns with a prior study that reported a reduction in cell growth and promotion of apoptosis in colorectal cancer cells through caspase-3 activation by *S. aromaticum* [51,54]. Markedly, at 200 µg/mL, SA induced the highest caspase-3/7 activity in HCT116 cells, with lower responses in p53-mutant HT-29 cells. Together with Annexin V staining, these results support the conclusion that both extracts activate apoptotic pathways, particularly in p53-competent cells. This p53 dependency was especially evident in the stronger pro-apoptotic effects of SA, further reinforcing its therapeutic promise.

To assess long-term antiproliferative effects, SA and SN extracts were evaluated using colony formation assays. This method evaluates the capacity of a single cell to proliferate and form a colony and is a widely accepted method for evaluating the long-term antiproliferative effects of anticancer agents [24,55]. Colony formation assays revealed that SA markedly reduced clonogenic survival in both cell lines, especially in HCT116, where colony numbers dropped to 11% at 200 µg/mL. This aligns with prior findings showing that *S. aromaticum* fractions suppress colony formation in colorectal cancer cells [56]. In contrast, SN extract exhibited only moderate inhibition in HCT116 cells and minimal effects in HT-29 cells across all concentrations. To explore the underlying basis of this difference, LC-DAD-MS/MS analysis revealed that SA contained higher levels of gallic acid than SN, which may contribute to its stronger antiproliferative effect. This observation is consistent with a previous report showing that gallic acid significantly reduced colony formation in colon cancer cells [24]. Overall, these findings suggest that SA possesses superior antiproliferative activity compared to SN, particularly in p53 wild-type colorectal cancer cells. The results highlight the therapeutic potential of SA as a viable candidate for long-term growth suppression in colorectal cancer, especially in p53-proficient tumors.

To investigate the underlying molecular mechanisms in cell cycle regulation, flow cytometric analysis was conducted to determine the cell cycle distribution, while Western blotting was performed to examine the expression of cell cycle-associated proteins. Both methods offer complementary insights by capturing population-level changes and molecular mechanisms induced by the extracts. This integrated approach has been successfully employed in previous studies investigating cell cycle modulation in colorectal cancer cells [57,58]. In the present study, flow cytometry revealed an accumulation of cells in the sub-G1 phase in HCT116 cells following treatment with both extracts, suggesting the presence of DNA-fragmented cells. This finding was supported by Annexin V-FITC/PI staining, which showed increased populations of Annexin V-positive cells (Figure 4), consistent with the shift observed in the sub-G1 region. In HT-29 cells, sub-G1 increase occurred only at the highest concentration (200 µg/mL). These results agree with previous studies demonstrating that *S. aromaticum* ethyl acetate extract induces G0/G1-phase arrest and triggers apoptotic cell death in colorectal cancer cells [11]. In addition, 2′,4′-Dihydroxy-6′-methoxy-3′,5′-dimethylchalcone (DMC), a bioactive compound purified from *S. nervosum* seeds, was shown to cause G0/G1-phase arrest in HeLa cells, as reported in previous studies [29].

Western blot results supported these findings by showing increased p21 and decreased cyclin D1 expression in HCT116; in particular, SA showed a higher intensity than SN treatment. These changes suggest G1/S arrest via p53–p21 signaling, consistent with previous colorectal cancer cell studies [59,60]. In contrast, HT-29 cells exhibited minimal or even opposing responses, highlighting the compromised p53-dependent regulatory mechanisms. Importantly, SA induced more robust alterations than SN, reinforcing its superior capacity to modulate cell cycle checkpoints. This is further supported by earlier reports showing that polyphenol-derived compounds from medicinal plants modulate key signaling cascades in cell cycles and apoptosis [61,62].

In summary, the apoptosis and antiproliferative findings offer valuable insights into the distinct mechanisms by which SA and SN extracts exert their anticancer effects. Importantly, SA extract modulates both cell proliferation and cell death pathways, highlighting its potential as a multifaceted therapeutic agent for colorectal cancer, especially in tumors retaining functional p53.

## 4. Materials and Methods

### 4.1. Plant Sample Preparation and Raw Material Quality Control

#### 4.1.1. Plant Identification

*S. aromaticum* was obtained from local crude drug markets in Chiang Mai, Thailand. The plant material was authenticated by comparison with authenticated specimens available in a botanical herbarium and referenced monographs. The samples were subsequently cleaned and dried in a hot air oven at a temperature of 50 °C. The fruits of *S. nervosum* were collected from Chiang Mai, Thailand. The sample was authenticated by Wannaree Charoensup, a botanist at the Faculty of Pharmacy, Chiang Mai University. A voucher specimen was deposited at the Laboratory of Pharmacognosy, Faculty of Pharmacy, Chiang Mai University, under the crude drug number 001/2023.

#### 4.1.2. Plant Extraction

Dried plant materials were ground and homogenized prior to extraction. For *S. aromaticum* (SA), the initial extraction was carried out using a Soxhlet apparatus with ethyl acetate as the solvent. Following this step, the residual clove hydrodistillation marc was subjected to further extraction with 95% ethanol. The extracts were filtered with Whatman No. 1 and concentrated under a vacuum using a rotary evaporator to yield ethyl acetate and ethanol-derived extracts of clove hydrodistillation residue, respectively. The latter was referred to as the SA extract for subsequent experiments. For *S. nervosum* (SN), the dried seeds were extracted by maceration in 95% ethanol for three days at room temperature. After the initial filtration using Whatman No. 1 filter paper, the plant residue was extracted again with 95% ethanol until exhaustion. To obtain the final SN seed extract, the combined ethanolic fractions were concentrated using a rotary evaporator (BUCHI Rotavapor^®^ R-300, Flawil, Switzerland) under vacuum.

### 4.2. Quality Control of the Extracts

#### 4.2.1. Identification and Quantification of Major Phytochemicals Using LC-DAD-Q-Orbitrap-MS/MS

Gallic acid was purchased from Tokyo Chemical Industry (TCI, Tokyo, Japan), and ellagic acid was obtained from Sigma-Aldrich (St. Louis, MO, USA). HPLC-grade methanol and LC-MS-grade formic acid were purchased from Thermo Fisher Scientific (Thermo Fisher Scientific, Pittsburgh, PA, USA). Deionized water was purified using a Cascada I Ultrapure Water System (Pall Corporation, Port Washington, NY, USA). All chemicals and solvents used were of analytical grade.

The LC-MS/MS analysis was conducted using a Vanquish UHPLC system (Thermo Fisher Scientific, Waltham, MA, USA) equipped with a Binary Pump F, Split Sampler FT, Column Compartment H, and a Diode Array Detector FG, all from Thermo Scientific, Waltham, MA, USA. The separation was conducted using a Hypersil BDS C18 column (100 × 2.1 mm i.d., 2.4 µm). The mobile phases were (A) 0.1% formic acid in water and (B) methanol. A mobile-phase time program was set up at a constant flow rate of 0.6 mL/min with 100% A for 3 min, then linear gradient elution from 0% to 40% B in A for 5 min, 40% B in A for 2 min, gradient from 40% to 90% B in A for 6 min, and 100% B for 4 min. Prior to each injection, the column was equilibrated with 100% A for 3 min. Column temperature was fixed at 25 °C with still air. Diode array detection was set at wavelengths of 254 and 272 nm. The injection volume was 2 µL for all samples and standards.

Mass spectrometric analysis was conducted in both positive and negative ionization modes, utilizing EASY-IC^TM^ internal mass calibration. The ion source employed was a Heated Electrospray Ionization (HESI) source. Spray voltage was set in the static mode at 3500 V for positive ions and 2500 V for negative ions. Nitrogen gas was supplied in static mode with flow rates configured as follows: sheath gas at 60 arbitrary units (Arb), auxiliary gas at 15 Arb, and sweep gas at 2 Arb. The ion transfer tube and vaporizer temperatures were both maintained at 350 °C. Full-scan data were acquired over a mass range of 100–1000 *m*/*z* at a resolution of 60,000, with the RF lens set to 70%. Data-dependent MS/MS (ddMS2) acquisition was enabled with an intensity threshold of 5.0 × 10^5^. MS^2^ settings included an isolation window of 1.5 *m*/*z*, normalized collision energy, Orbitrap resolution of 15,000, and automatic scan range mode. Instrument operation and data acquisition were controlled using the Chromeleon™ Chromatography Data System software (version 7.3). Chromatograms were visualized and analyzed using FreeStyle software (version 1.8.65.0) Compound identification was performed by comparing acquired mass spectral data with reference spectra from the National Institute of Standards and Technology (NIST) and mzCloud databases, as well as previously published literature.

Quantification was based on the peak area obtained from a diode array detector. A stock solution of gallic acid was prepared by precisely weighing the compound and dissolving it in a 1:1 (*v*/*v*) methanol–water mixture to achieve a final concentration of 2000 µg/mL. In parallel, ellagic acid stock solution was freshly prepared by dissolving the accurately weighed compound in 0.01 N sodium hydroxide. Working standard solutions for both compounds were prepared by serial dilution of the stock solutions using the same methanol–water solvent system. Calibration curves were constructed across a concentration range of 3.9–1000 µg/mL by plotting the peak area against the corresponding concentrations to assess linearity. For sample preparation, *S. aromaticam* was accurately weighed and extracted with a methanol–water mixture (1:1, *v*/*v*) to yield a concentration of 5 mg/mL. The extraction was carried out in an ultrasonic bath for 30 min. A similar method was applied for *S. vernosam*, with 0.01 N sodium hydroxide as the extraction solvent. All samples were completed in triplicate to ensure reproducibility. Prior to HPLC injection, each extract was filtered through a 0.2 µm nylon membrane to remove particulates and ensure sample clarity.

#### 4.2.2. Quantification of Total Phenolic Compounds Using the Folin–Ciocalteu Assay

The total phenolic compounds in both SA and SN extracts were assessed using the Folin–Ciocalteu assay [63]. Plant extracts were solubilized in DMSO at a concentration of 1 mg/mL. The assay was performed in a 96-well microplate format where 20 µL of each extract solution was combined with 100 µL of freshly prepared Folin–Ciocalteu reagent. This mixture was then supplemented with 80 µL of sodium carbonate solution (prepared at 75 g/L) to create an alkaline environment necessary for the reaction. After mixing thoroughly, the plate was incubated at an ambient temperature for a period of 30 min to allow complete color development. Following incubation, we measured the absorbance of each well at 765 nm using a spectrophotometric microplate reader (BioTek, Winooski, VT, USA). To enable quantification, we established a standard calibration curve using varying concentrations of gallic acid analyzed under identical conditions. Results were calculated and expressed as milligrams of gallic acid equivalents (GAE) per gram of dry extract weight, allowing for a standardized comparison between samples.

#### 4.2.3. Evaluating Potential of Plant Extracts Using Antioxidant Assays

The free radical scavenging activity of SA and SN extracts was determined using a DPPH assay adapted from the method published by Vongsak et al. [64]. This protocol involved mixing 100 μL of each extract with 100 μL of 152 μM DPPH solution prepared in methanol. After combining these components, we allowed the mixtures to react for 30 min at room temperature under dark conditions. Using a microplate reader (BioTek Instruments, Winooski, VT, USA), we then measured the absorbance at 517 nm for each sample. As a reference, we prepared a negative control consisting of methanol-DPPH solution without any extract. We calculated the radical scavenging activity as the percent inhibition using the formula % Inhibition = [(A__control_ − A__sample_)/A__control_] × 100, where A_control_ is the absorbance of the DPPH solution without the sample and A__sample_ is the absorbance in the presence of the extract. To ensure reliability, we performed all measurements in triplicate and reported the average values.

The antioxidant properties of SA and SN extracts were assessed using a modified ABTS radical scavenging assay, adapted from the protocol described by Arnao et al. [65]. To prepare the ABTS^+•^ stock solution, equal volumes of 7.4 mM ABTS and 2.6 mM potassium persulfate were combined and stored in darkness at room temperature for 12–16 h to generate the radical cation. Before testing, the ABTS^+•^ solution was diluted with sodium phosphate buffer (pH 7.0) until reaching an absorbance of approximately 1.1 ± 0.02 at 750 nm, as measured by UV-vis spectrophotometry. For the assay, 20 μL of each appropriately diluted extract sample was combined with 200 μL of the prepared ABTS^+•^ solution in wells of a 96-well microplate. Following a 5 min incubation period at room temperature in darkness, absorbance measurements were taken at 750 nm using a microplate reader (BioTek Instruments, Winooski, VT, USA). Vitamin C was used as a positive control, and IC_50_ values were calculated from serial dilutions of extract concentrations. All measurements were performed in triplicate to ensure reliability.

### 4.3. In Vitro Assay

#### 4.3.1. Cell Culture

This study utilized two colorectal cancer cell lines: the p53 wild-type human colorectal cancer cell line (HCT116), and the p53 mutant human colorectal cancer cell line (HT-29). These cell lines were provided by Associate Professor Dr. Teera Chewonarin from the Department of Biochemistry, Faculty of Medicine, Chiang Mai University and purchased from ATCC (Manassas, VA, USA). The cells were continuously cultured in filter culture flasks (The University of Nottingham Ningbo (NUNC), China) with DMEM (Corning, NY, USA) supplemented with 10% fetal bovine serum (Gibco™ Fetal Bovine Serum; Thermo Fisher Scientific, Waltham, MA, USA) in a CO_2_ incubator (Thermo Fisher Scientific (Langenselbold Facility): Langenselbold, Germany) under a humidified atmosphere of 5% CO_2_ at 37 °C. When the cell density reached approximately 70–80% confluence, we harvested the cells for experimental use in further experiments, cryopreservation, or routine passaging.

#### 4.3.2. Cell Viability Assay

HCT116 and HT-29 cells were seeded in 96-well plates for 24 h at a density of 5 × 10^3^ cells per deepening and exposed to both SA and SN extracts for an additional 24 h. The cell lifespan was determined using an MTT assay (Sunshinebio, Nanjing, China). Twenty microliters (20 RAW) of MTT (5 mg/mL) were administered for each deepening, and microplates were incubated with 5% CO at 37 °C for 4 h. The optical density (OD) of each sample was recognized at 540 nm/630 nm using an automated record reader (Organic Wheel, Hercules, CA, USA).

#### 4.3.3. Apoptosis Assay

Cell apoptosis was observed using the Annexin V-FITC/PI Apoptosis Detection Kit (Thermo Fisher Scientific, Waltham, MA, USA) according to the manufacturer’s instructions. HCT116 and HT-29 cells were plated at a density of 5 × 10^4^ cells in 24-well plates. After treatment with both SA and SN extracts, the cells were washed twice with PBS and resuspended in 500 µL of binding buffer at a density of 1 × 10^6^ cells/mL. Then, 5 µL of Annexin V-FITC (excitation/emission at λ = 488 nm/525 nm) was added to the cell suspension, followed by 5 µL of propidium iodide (PI, excitation/emission at λ = 535 nm/617 nm). After incubating for 15 min, the samples were assessed using a CytoFLEX flow cytometer (Beckman Coulter, La Brea, CA, USA).

#### 4.3.4. Caspase 3/7 In Situ Fluorescence Microscope

HCT116 and HT-29 cells were plated at a density of 5 × 10^5^ cells following treatment with both SA and SN extracts. The CellEvent™ Caspase-3/7 Detection Reagents (Thermo Fisher Scientific, Waltham, MA, USA) were used to stain the cells according to the manufacturer’s protocol. Fluorescence microscopy (Zeiss Microscopes, Jena, Germany) was performed immediately to analyze the samples, and ZEN 3.5 (Blue Edition) Microscopy Software was used to measure fluorescence intensity.

#### 4.3.5. Colony Formation Assay

HCT116 and HT-29 cells (500 cells per well) were seeded in 6-well plates and permitted to attach overnight. The cells were then treated with increasing concentrations of SA and SN extracts (0, 25, 50, 100, and 200 µg/mL) for 24 h. After treatment, the medium was replaced with fresh DMEM containing 10% FBS, and cells were incubated at 37 °C for 14 days. Colonies were fixed with methanol and stained with 0.5% crystal violet. Colonies containing more than 50 cells were quantified using ImageJ software. Data were normalized to the control group and analyzed using one-way ANOVA, with *p* < 0.05 considered statistically significant.

#### 4.3.6. Cell Cycle Assay

HCT116 and HT-29 cells were seeded into 6-well plates and incubated for 24 h prior to treatment with varying concentrations of SA and SN extracts. At 24 and 48 h post-treatment, cells were harvested, fixed in 70% ethanol overnight, and stained with propidium iodide (PI) for cell cycle analysis. Phase distribution was analyzed using a BD FACScan™ flow cytometer (BD Biosciences, San Jose, CA, USA). All measurements were performed in triplicate across three independent experiments [66].

#### 4.3.7. Western Blotting

After treating with SA and SN extracts, HCT116 and HT-29 cells were lysed in RIPA buffer supplemented with a protease inhibitor cocktail tablet (Roche, Mannheim, Germany). Protein concentrations were determined using the Bradford assay, as described by Subhawa et al. [67]. Equal amounts of total cellular protein were subjected to SDS-polyacrylamide gel electrophoresis (SDS-PAGE) and subsequently transferred onto nitrocellulose membranes. To minimize non-specific antibody binding, membranes were blocked with 5% skim milk dissolved in 0.1% Tween-20 in TBS for 1 h at ambient temperature. Membranes were incubated overnight at 4 °C with primary antibodies specific to apoptosis-related proteins, including p21 and cyclin D1 (Cell Signaling Technology, Danvers, MA, USA). β-actin was used as a reference loading control (Sigma-Aldrich, St. Louis, MO, USA). After washing, membranes were incubated with HRP-conjugated secondary antibodies (anti-rabbit IgG and anti-mouse IgG; Thermo Fisher Scientific, Waltham, MA, USA) for 2 h at room temperature. Protein bands were detected using the ImmunoStar^®^ Zeta chemiluminescent substrate (Fujifilm Wako Pure Chemical Corporation, Osaka, Japan). The relative intensity of each band was quantified using ImageJ software version 1.54 g (NIH, Bethesda, MD, USA).

### 4.4. Statistical Analysis

Data are expressed as mean ± standard deviation (SD). The one-way ANOVA with Tukey’s post hoc test was used to conduct all statistical analyses. Statistical significance was set at *p* < 0.05 and analysis was performed using IBM SPSS Statistics, version 22.0 (International Business Machines Corporation, Armonk, NY, USA).

## 5. Conclusions

This study highlights the potential of clove hydrodistillation residue extract from *Syzygium aromaticum* (SA) and seed extract from *Syzygium nervosum* (SN) as sources of natural polyphenols, particularly gallic acid and ellagic acid. Both extracts exhibited antioxidant activity and exerted cytotoxic effects against HCT116 colorectal cancer cells through induction of apoptosis and cell cycle arrest. These effects were more pronounced in wild-type p53 cells, whereas limited responses were observed in mutant p53-bearing HT-29 cells, suggesting a p53-dependent mechanism. Importantly, both extracts demonstrated preferential cytotoxicity toward cancer cells over normal fibroblasts, underscoring their potential selectivity and safety. While these in vitro findings provide a promising foundation, further in vivo investigations are needed to confirm efficacy, assess systemic toxicity, and elucidate molecular mechanisms using relevant biomarkers such as gene expression profiles and DNA damage indicators. In particular, comparative studies with standard chemotherapeutic agents in animal models should be conducted to determine the relative therapeutic efficacy of these extracts within a translational context. Furthermore, advanced delivery platforms, such as nanoparticle-based systems, may enhance bioavailability and improve therapeutic outcomes, especially in p53-deficient or drug-resistant colorectal cancers. Establishing such preclinical evidence will be critical to support future clinical translation.

## Figures and Tables

**Figure 1 ijms-26-06826-f001:**
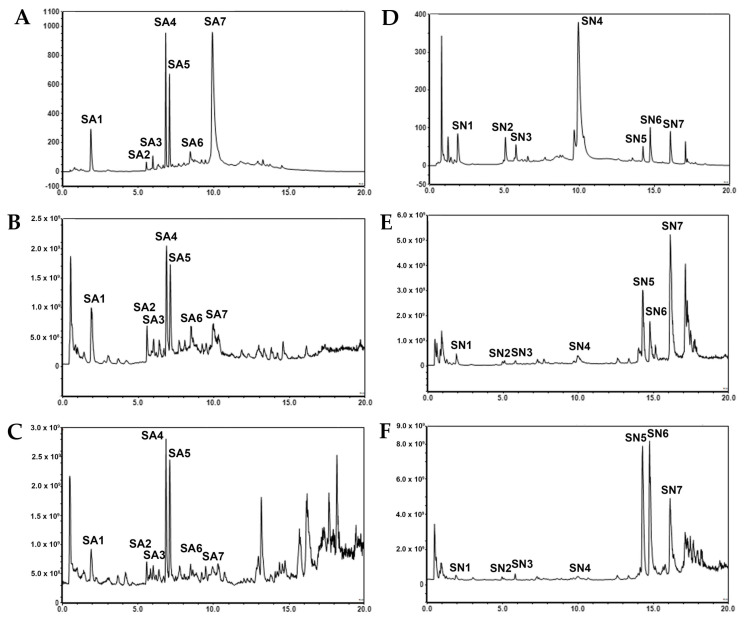
Chromatograms of *S. aromaticum* flower bud waste extract detected with (**A**) UV 254 nm, (**B**) total ion counts of negative mode with full scan at 100–1000 *m*/*z*, and (**C**) total ion counts in positive mode with full scan at 100–1000 *m*/*z*. Chromatograms of *S. nervosum* extract detected with (**D**) UV 254 nm, (**E**) total ion counts in negative mode with full scan at 100–1000 *m*/*z*, and (**F**) total ion counts in positive mode with full scan at 100–1000 *m*/*z*.

**Figure 2 ijms-26-06826-f002:**
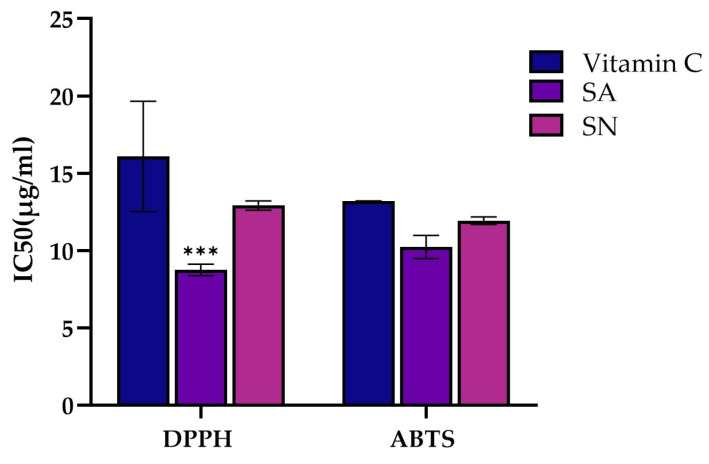
Comparative antioxidant activities of vitamin C, *Syzygium aromaticum* (SA), and *Syzygium nervosum* (SN) extracts as measured by DPPH and ABTS radical scavenging assays. In the DPPH assay, SA exhibited significantly stronger antioxidant activity compared to vitamin C. Data are presented as mean ± SD (*n* = 3 independent experiments). Statistical analysis was performed using one-way ANOVA followed by Tukey’s post hoc test. Statistical significance versus control (vitamin C): *** *p* < 0.001.

**Figure 3 ijms-26-06826-f003:**
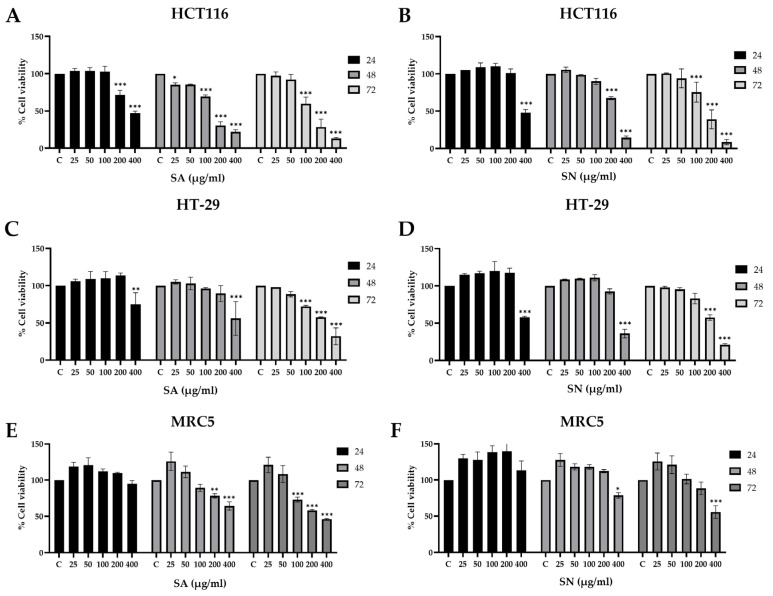
Cytotoxic responses of colorectal cancer cells to SA and SN extracts in a dose- and time-dependent manner. HCT116 (**A**,**B**), HT-29 (**C**,**D**), and MRC-5 (**E**,**F**) cells were exposed to varying concentrations of SA and SN extracts for 24, 48, and 72 h, followed by evaluation of cell viability using the MTT assay. Data are expressed as mean ± SD (*n* = 3). The data represents three independent experiments. Statistical significance vs. control: * *p* < 0.05, ** *p* < 0.01, and *** *p* < 0.001.

**Figure 4 ijms-26-06826-f004:**
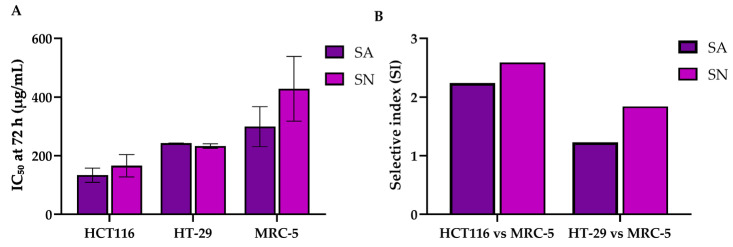
Quantitative plot of cytotoxic responses of colorectal cancer cells to SA and SN extracts. (**A**) Comparison of IC_50_ values of SA and SN extracts at 72 h across HCT116 (p53-wild-type), HT-29 (p53-mutant), and MRC-5 (normal fibroblast) cell lines. (**B**) Selectivity index (SI), calculated as the ratio of IC_50_ in MRC-5 to that in CRC cells. Data are expressed as mean ± SD from three independent experiments (*n* = 3).

**Figure 5 ijms-26-06826-f005:**
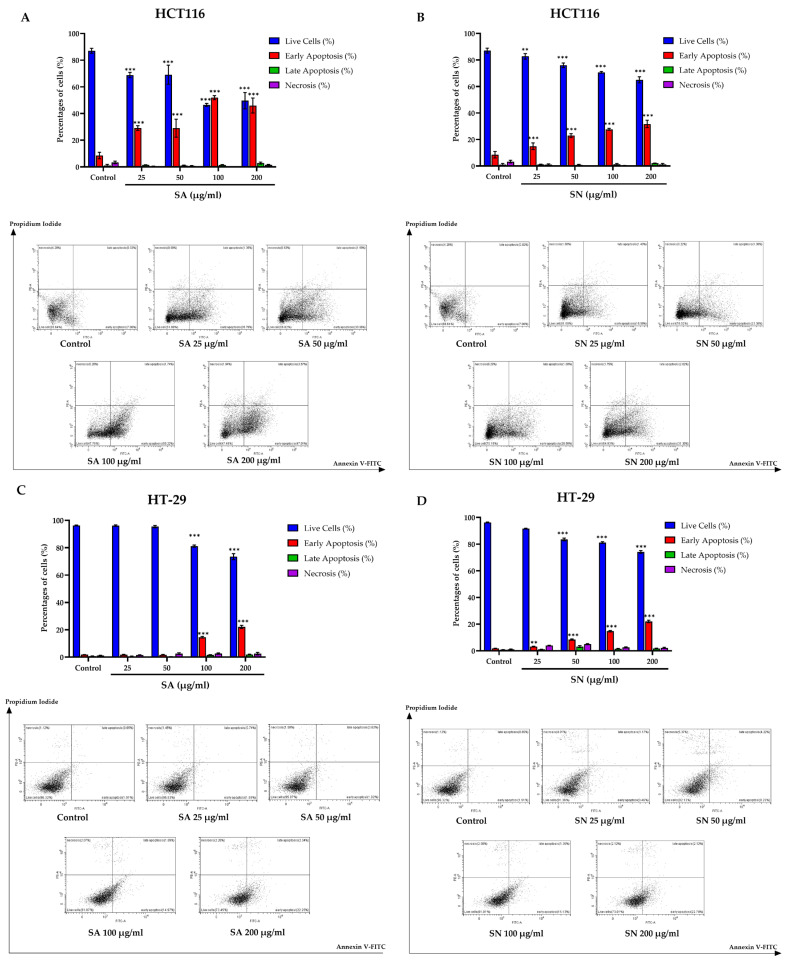
Flow cytometric analysis of apoptosis in HCT116 and HT-29 cells following treatment with SA and SN extracts. Annexin V-FITC and propidium iodide staining were performed after 24 h of treatment, followed by flow cytometric analysis. (**A**,**B**) HCT116 cells treated with SA and SN extracts; (**C**,**D**) HT-29 cells treated with SA and SN extracts. Bar graphs represent the percentages of live (blue), early apoptotic (red), late apoptotic (green), and necrotic (purple) cells. Representative dot plots are shown below each bar graph. Data are expressed as mean ± SD from three independent experiments. *p* values indicate statistical significance compared to untreated control; ** *p* < 0.01, and *** *p* < 0.001.

**Figure 6 ijms-26-06826-f006:**
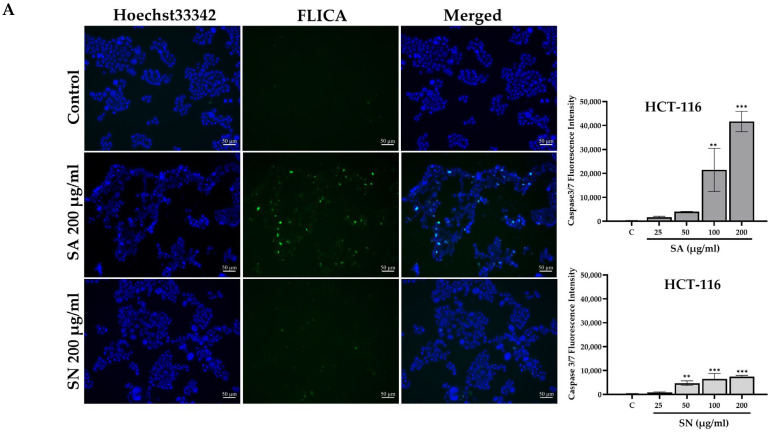
Caspase-3/7 activity induced by SA and SN extracts in colorectal cancer cells. (**A**) Representative fluorescence images showing caspase-3/7 activation in HCT116 cells treated with 200 µg/mL of SA and SN extracts for 24 h. Nuclei were visualized using Hoechst 33342 (blue), while active caspase-3/7 was detected using a FLICA reagent (green). Merged images demonstrate the co-localization of apoptotic signals with nuclear staining. Scale bars = 50 µm. The corresponding bar graph shows a dose-dependent increase in caspase-3/7 fluorescence intensity in HCT116 cells. (**B**) Similar analysis was performed in HT-29 cells exposed to the same extracts and concentrations. Quantitative data are presented as mean ± SD from three independent experiments. Statistical significance compared to control; * *p* < 0.05, ** *p* < 0.01, and *** *p* < 0.001.

**Figure 7 ijms-26-06826-f007:**
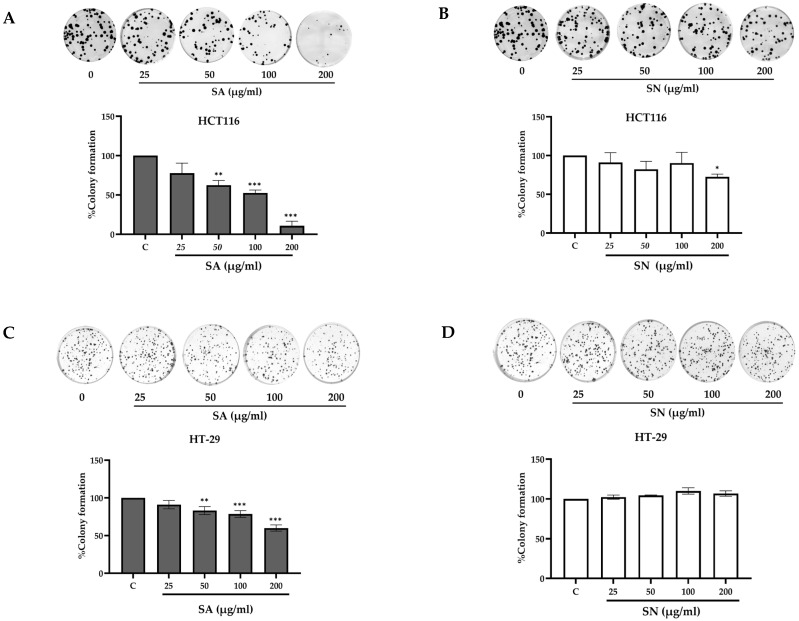
Inhibitory effects of SA and SN extracts on colony formation in colorectal cancer cells. (**A**,**B**) HCT116 and (**C**,**D**) HT-29 cells were exposed to increasing concentrations of SA and SN extracts (25–200 µg/mL) for 24 h. Colonies were fixed, stained, and quantified. Bar graphs represent the percentage of colony formation relative to untreated controls. Representative images of colony plates are shown above the graphs. Data are expressed as mean ± SD from three independent experiments. Statistical analysis was performed using one-way ANOVA with post hoc test; * *p* < 0.05, ** *p* < 0.01, and *** *p* < 0.001 vs. control.

**Figure 8 ijms-26-06826-f008:**
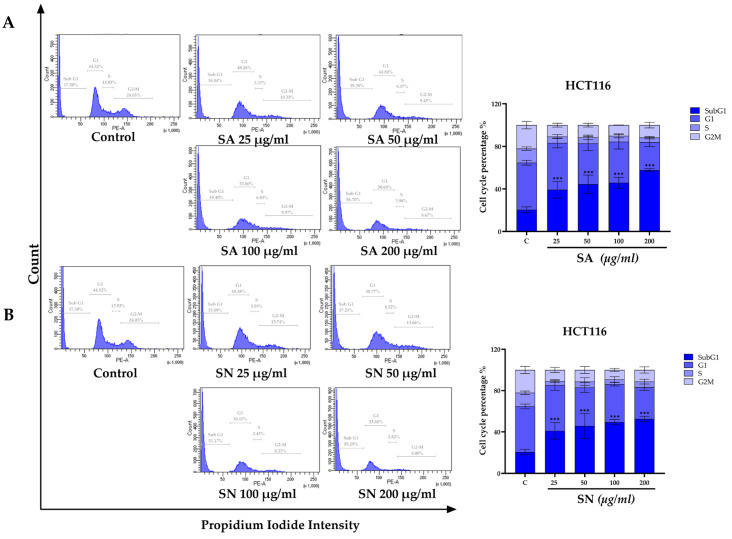
Analysis of cell cycle distribution in colorectal cancer cells treated with SA and SN extracts. (**A**,**B**) HCT116 and (**C**,**D**) HT-29 cells were treated with increasing concentrations (25–200 µg/mL) of SA and SN for 24 h. Flow cytometry of propidium iodide-stained cells was used to evaluate DNA content. Cell cycle phase distribution (sub-G1, G1, S, and G2/M) is depicted in the bar charts as percentage values. Data are expressed as mean ± SD from three independent experiments. Statistical significance vs. control; and *** *p* < 0.001.

**Figure 9 ijms-26-06826-f009:**
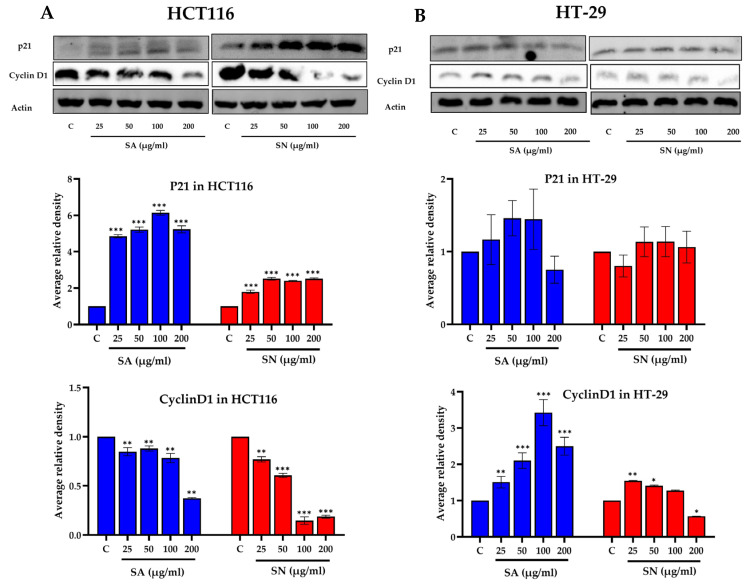
Effects of *S. aromaticum* (SA) and *S. nervosum* (SN) extracts on the expression of cell cycle-related proteins in colorectal cancer cells. (**A**) HCT116 cells were treated with increasing concentrations (25–200 µg/mL) of SA or SN extracts for 24 h. Western blotting was performed to detect the expression of p21 and cyclin D1, with β-actin used as a loading control. The bar graphs show a densitometric quantification of band intensities normalized to β-actin and expressed relative to the untreated control. (**B**) HT-29 cells were subjected to the same treatments and analyses as described in panel A. Bar graphs represent a densitometric quantification of protein bands relative to control using ImageJ software. Data are shown as mean ± SD with n = 3. Statistical significance was determined using one-way ANOVA with post hoc test: * *p* < 0.05, ** *p* < 0.01, and *** *p* < 0.001 vs. control.

**Table 1 ijms-26-06826-t001:** Identification of major components in *S. aromaticum* (SA) extract sample.

Peak No.	Retention Time (min)	Compound	Mode	Parent Ion (MS^1^)	Fragment Ion (MS^2^)
SA1	1.91	Gallic acid C_7_H_6_O_5_, MW = 170	Negative	169.0140 (Calc for [M−H]^−^ = 169.01315)	125.0245
			Positive	171.0288 (Calc for [M+H]^+^ = 171.02880)	153.0183, 127.0390, 125.0234, 109.0284, 107.0127, 81.0335
SA2	5.58	Unknown C_20_H_20_O_14_ MW = 484	Negative	483.0773 (Calc for [M−H]^−^ = 483.07693)	331.0677, 313.0566, 169.0142, 125.0245
			Positive	485.0925 (Calc for [M+H]^+^ = 485.09258)	202.0779, 153.0182
				507.0744 (Calc for [M+Na]^+^ = 507.07453)	337.0526, 202.0778, 153.0185
SA3	6.02	Stictinin C_27_H_22_O_18_ MW = 634	Negative	633.0729 (Calc for [M−H]^−^ = 633.07224)	300.9988, 275.0199, 202.0787, 169.0141
			Positive	657.0695 (Calc for [M+Na]^+^ = 657.06983)	N/A
SA4	6.87	Biflorin or isobiflorin C_16_H_18_O_9_ MW = 354	Negative	353.0870 (Calc for [M−H]^−^ = 353.08671)	233.0456, 205.0506, 202.0789
			Positive	355.1019 (Calc for [M+H]^+^ = 355.10236)	337.0921, 319.0810, 301.0715, 289.0709, 273.0757, 259.0605, 245.0802, 235.0603, 205.0498, 202.0780
SA5	7.11	Biflorin or isobiflorin C_16_H_18_O_9_ MW = 354	Negative	353.0869 (Calc for [M−H]^−^ = 353.08671)	263.0563, 245.0458, 233.0455, 205.0506, 202.0790
			Positive	355.1020 (Calc for [M+H]^+^ = 355.10236)	337.0906, 319.0818, 301.0713, 283.0608, 271.0605, 259.0602, 245.0805, 235.0602, 231.0653, 205.0496, 202.0779
SA6	8.47	Unknown C_13_H_8_O_7_ MW = 276	Negative	247.0247	219.0298, 202.0789, 191.0350
				275.0196 (Calc for [M−H]^−^ = 275.01863)	257.0090, 229.0142, 202.0788
			Positive	249.1119	207.0287, 202.0778
				277.0342 (Calc for [M+H]^+^ = 277.03428)	259.0236, 231.0287, 215.0338, 202.0778, 187.0390
SA7	9.97	Ellagic acid C_14_H_6_O_8_ MW = 302	Negative	300.9986 (Calc for [M−H]^−^ = 300.99789)	283.9955, 257.0091, 245.0097, 229.0150, 202.0789, 185.0245
			Positive	303.0134 (Calc for [M+H]^+^ = 303.01354)	285.0031, 275.0186, 257.0082, 202.0777
				324.9954 (Calc for [M+Na]^+^ = 344.99549)	202.0777

**Table 2 ijms-26-06826-t002:** Identification of major components in *S. nervosum* (SN) extract sample.

Peak No.	Retention Time (min)	Compound	Mode	Parent Ion (MS^1^)	Fragment Ion (MS^2^)
SN1	1.90	Gallic acid C_7_H_6_O_5_ MW = 170	Negative	169.0141 (Calc for [M−H]^−^ = 169.01315)	125.0244
			Positive	171.0282 (Calc for [M+H]^+^ = 171.02880)	153.0177, 135.0073, 127.0385, 125.0229, 109.0280, 107.0123, 81.0332
SN2	5.10	Pedunculagin or its isomer C_34_H_24_O_22_ MW = 784	Negative	391.0303 (Calc for [M−2H]^2−^ = 391.03068)	300.9984, 275.0197, 202.0786
				783.0677 (Calc for [M−H]^−^ = 783.06755)	300.9988, 275.0197
			Positive	785.0827 (Calc for [M+H]^+^ = 785.08320)	N/A
				799.0986	303.0134, 276.0264
				802.1094 (Calc for [M+NH_4_]^+^ = 802.10975)	303.0133, 277.0341, 259.0237
SN3	5.84	Pedunculagin or its isomer C_34_H_24_O_22_ MW = 784	Negative	391.0302 (Calc for [M−2H]^2−^ = 391.03068)	300.9985, 275.0196, 202.0788
				783.0677 (Calc for [M−H]^−^ = 783.06755)	300.9989, 275.0197
			Positive	243.0473	202.0777, 203.0853
				463.1057	243.0476, 202.0777, 203.0847
				802.1097 (Calc for [M+NH_4_]^+^ = 802.10975)	N/A
SN4	9.96	Ellagic acid C_14_H_6_O_8_ MW = 302	Negative	300.9987 (Calc for [M−H]^−^ = 300.99789)	283.9963, 202.0789
			Positive	303.0134 (Calc for [M+H]^+^ = 303.01354)	285.0031, 275.0188, 257.0082
				324.9954 (Calc for [M+Na]^+^ = 344.99549)	202.0777, 203.0843, 227.9203
SN5	14.28	Unknown Dimer of 2′,4′-Dihydroxy-6′-methoxy-3′,5′-dimethylchalcone C_36_H_36_O_8_ MW = 596	Negative	297.1129 (Calc for [M−2H]^2−^ = 297.11323)	282.0896, 255.1027, 240.0792, 202.0788, 193.0507, 178.0272, 150.0322
			Positive	299.1275 (Calc for [M+2H]^2+^ = 299.12779)	284.1057, 202.0777, 195.0652
				321.1093	306.0862, 202.0778, 203.0846
				619.2296 (Calc for [M+Na]^+^ = 619.23024)	321.1097, 202.0780, 203.0837
SN6	14.76	Unknown Dimer of 2′,4′-Dihydroxy-6′-methoxy-3′,5′-dimethylchalcone C_36_H_36_O_8_ MW = 596	Negative	297.1130 (Calc for [M−2H]^2−^ = 297.11323)	255.1028, 211.1129, 202.0789, 193.0507, 149.0609, 79.0190
				617.2155	297.1131, 255.1028, 193.0506
			Positive	299.1275 (Calc for [M+2H]^2+^ = 299.12779)	213.0758, 195.0652
				321.1095	202.0776, 203.0847, 217.0467, 233.3068
				619.2299 (Calc for [M+Na]^+^ = 619.23024)	321.1097, 202.0780, 203.0837
SN7	16.10	2′,4′-Dihydroxy-6′-methoxy-3′,5′-dimethylchalcone (DMC) C_18_H_18_O_4_ MW = 298	Negative	297.1127 (Calc for [M^−^H]^−^ = 297.11323)	253.1234, 202.0789, 193.0506
			Positive	299.1275 (Calc for [M+H]^+^ = 299.12779)	213.0757, 195.0652
				321.1094 (Calc for [M+Na]^+^ = 321.10973)	202.0778, 203.0840, 235.0583

**Table 3 ijms-26-06826-t003:** IC_50_ values (µg/mL) of SA and SN extracts in HCT116, HT-29, and MRC-5 cell lines at 24, 48, and 72 h post-treatment, and the corresponding Selectivity Index (SI) at 72 h.

Compounds/Time	Cell Lines		
	HCT116	HT-29	MRC-5
***Syzygium aromaticum*** **(SA)**	**IC_50_ values (µg/mL)**		
**24 h**	352.85 ± 51.40	N/A	N/A
**48 h**	140.11 ± 5.89	346.50 ± 14.19	N/A
**72 h**	133.65 ± 24.01	242.94 ± 0.70	299.3 ± 68.2
**SI ***	2.24	1.23	-
***Syzygium nervosum*** **(SN)**	**IC_50_ values (µg/mL)**		
**24 h**	386.99 ± 4.97	N/A	N/A
**48 h**	265.01 ± 12.26	354.54 ± 17.50	N/A
**72 h**	165.64 ± 37.79	232.92 ± 8.08	428.3 ± 110.6
**SI ***	2.59	1.84	-

Data are presented as mean ± SD (*n* = 3). IC_50_ values were calculated from dose–response curves using nonlinear regression analysis. N/A indicates that the IC_50_ value could not be determined within the tested concentration range. * Selectivity Index (SI) values were calculated as the ratio of IC_50_ in MRC-5 cells to IC_50_ in cancer cells (HCT116 or HT-29) at 72 h.

**Table 4 ijms-26-06826-t004:** Colony-forming efficiency.

Extract Compounds	Colony-Forming Efficiency (%)				
	Control	25 µg/mL	50 µg/mL	100 µg/mL	200 µg/mL
	**HCT116**				
***Syzygium aromaticum*** **(SA)**	100.00 ± 0	77.67 ± 12.73	62.44 ± 5.98 **	52.61 ± 3.69 ***	10.80 ± 5.84 ***
***Syzygium nervosum*** **(SN)**	100.00 ± 0	91.05 ± 12.66	82.28 ± 10.29	90.30 ± 13.72	72.58 ± 3.43 *
	**HT-29**				
***Syzygium aromaticum*** **(SA)**	100.00 ± 0	91.01 ± 5.53	83.21 ± 5.25 **	78.71 ± 4.43 ***	60.05 ± 4.11 ***
***Syzygium nervosum*** **(SN)**	100.00 ± 0	102.30 ± 2.59	104.50 ± 0.49	110.00 ± 4.02	106.80 ± 3.41

Data are presented as mean ± SD (*n* = 3). Statistical analysis was performed using one-way ANOVA, followed by a post hoc test comparing each treatment to the control group. * *p* < 0.05, ** *p* < 0.01, and *** *p* < 0.001.

## Data Availability

The data from this study can be provided on request by the corresponding author.

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
