# Peer review of "The Effect of the Ethanolic Extracts from Syzygium aromaticum and Syzygium nervosum on Antiproliferative Activity and Apoptosis in HCT116 and HT-29 Cells"

_ijms, 2025, doi:10.3390/ijms26146826_

Round 1
Reviewer 1 Report
Comments and Suggestions for Authors
Please check the correct spelling of scientific names
I read the article carefully and from the beginning I found it very interesting. Colon cancer is an increasingly common global problem, even in places where it was not common. This is supposed to be due to changes in diet.
Syzygium aromaticum and S. nervosum are two species of aromatic plants; above all, S. aromaticum is widely used as a condiment.
The work consists of the following key points:
Identification and quantification of phenolic compounds.
The analytical methods are adequate and were able to identify the phenolic compounds present in the extracts; gallic acid predominates in both species.
For the antioxidant capacity they used two techniques with which they demonstrated that both extracts have a strong antioxidant capacity.
Colorectal Cancer
Cell viability: both extracts showed cytotoxic effects in both cell lines, with SA being stronger.
The authors investigated whether the cytotoxic effect shown by the extracts was due to their induction of apoptosis. To do this, they performed Annexin Apoptosis Assay in which SA turned out to have a strong effect with respect to SN.
They also investigated whether the effect was mediated by caspases. The results indicate that both SA and SN extracts effectively induce apoptosis through caspase-3/7 activation in colorectal cancer cells, with a stronger effect detected in the HCT116 cells (p53 wild-type) than in the HT-29 cells (p53-mutant).
Inhibition of Cell Growth and Induction of Cell Cycle Arrest by SA and SN Extracts in Colorectal Cancer Cells
Colony formation
The SA extract inhibited the growth of colonies from the dose of 50 micrograms.
Cell Cycle Analysis by Flow Cytometry and Western Bloing
The antiproliferative effects of SA and SN extracts were associated with cell cycle disruption, SA and SN extracts in p53 wild-type HCT116 cells.
Authors conclusions: the apoptosis and antiproliferative data provide insight into the distinct mechanisms underlying the anticancer effects of SA and SN extracts. The SA extract modulated both cell cycle progression and apoptotic pathways, supporting its potential as a multifaceted therapeutic agent for colorectal cancer, particularly in tumors with functional p53.These conclusions are correct based on the experiments carried out.
Author Response
Dear Reviewer,
Thank you for the opportunity to review this manuscript. I sincerely appreciate the trust you have placed in me.
The submitted manuscript, entitled “Anticancer Activities of the Ethanolic Extracts from Syzygium aromaticum and Syzygium nervosum via Cell Cycle Arrest and Apoptosis in Colorectal Cancer Cells” by Thunyatorn Yimsoo et al., has been submitted for consideration as an article in the International Journal of Molecular Sciences.
In accordance with your recommendation, we have carefully addressed the reviewers’ comments and made corresponding revisions to the manuscript.
Below are my major comments and suggestions for further revision. Kindly consider the following points:
- Please check the correct spelling of scientific names
Response: We sincerely thank the reviewer for this valuable comment. In response, we have carefully reviewed the manuscript to ensure that all scientific names particularly Syzygium aromaticum and Syzygium nervosum—are spelled accurately and consistently throughout, following the standard conventions of scientific nomenclature. Revisions have been made accordingly.
We thank the reviewer again for this valuable suggestion, which will help shape and strengthen the direction of our future research.

Reviewer 2 Report
Comments and Suggestions for Authors
In this manuscript, Yimsoo, et al, evaluated the effects of extracts from Syzygium aromaticum (SA) and Syzygium nervosum (SN), particularly gallic acid and ellagic acid. Both extracts exhibited antioxidant activity and cytotoxic effects against HCT116 colorectal cancer cells, with evidence of apoptosis induction and cell cycle arrest. These effects were more pronounced in cells with wild-type p53, HTC116 cells compared to mutant p53-bearing HT-29 cells. SA exhibited better biological activity compared to SN. Standard methods were used and the data is well presented. However there are concerns that need to be addressed.
- Use of normal human colon cell lines as controls is missing.
- These results are based solely on in vitro experiments, further in vivo animal studies are needed to evaluate the efficacy and safety of these extracts.
- Use of words such as "upstream railway", "hyphenated" distracts the reader and appears using words not generally used in these type of scientific manuscripts.
Author Response
Dear Reviewer,
Thank you for the opportunity to review this manuscript. I sincerely appreciate the trust you have placed in me.
The submitted manuscript, entitled “Anticancer Activities of the Ethanolic Extracts from Syzygium aromaticum and Syzygium nervosum via Cell Cycle Arrest and Apoptosis in Colorectal Cancer Cells” by Thunyatorn Yimsoo et al., has been submitted for consideration as an article in the International Journal of Molecular Sciences.
In accordance with your recommendation, we have carefully addressed the reviewers’ comments and made corresponding revisions to the manuscript.
Below are my major comments and suggestions for further revision. Kindly consider the following points:
- Use of normal human colon cell lines as controls is missing.
Response: We sincerely appreciate the reviewer’s thoughtful comment. The primary objective of this study was to evaluate the initial anticancer potential of ethanolic extracts from Syzygium aromaticum and Syzygium nervosum in human colorectal cancer cells. We focused on key cancer-associated mechanisms, including cell proliferation, apoptosis, cell cycle progression, and the expression of related regulatory proteins.
This study specifically aimed to evaluate the anticancer efficacy of the extracts against two colorectal cancer cell lines, HCT116 (wild-type p53) and HT-29 (mutant p53). These cell lines were deliberately selected based on their distinct p53 statuses, allowing us to explore both p53-dependent and p53-independent mechanisms. Our preliminary results demonstrated substantial growth inhibition and induction of apoptosis in both models. Importantly, these molecular pathways are generally inactive or tightly regulated in normal cells, which do not exhibit aberrant proliferation or dysregulation of the p53 signaling cascade observed in cancer cells. Furthermore, many studies focusing on anticancer efficacy published in the International Journal of Molecular Sciences have also employed cancer cell lines without including normal cells, particularly when the primary objective is to dissect tumor-specific signaling pathways.
- https://www.mdpi.com/1422-0067/21/22/8692
- https://www.mdpi.com/1422-0067/20/23/5914
- https://www.mdpi.com/1422-0067/26/7/2903
- https://www.mdpi.com/1422-0067/24/1/617
- https://www.mdpi.com/1422-0067/24/19/14519
However, comprehensive toxicity assessments in normal human colon cell lines and in vivo safety profiling are actively underway as part of our follow-up preclinical investigations, which will be reported in subsequent publications. This phased approach ensures rigorous validation of both efficacy and safety before clinical translation. To reflect your suggestion, a statement has been added in the Conclusion section on Page 21-22, lines 884-888, indicating the need for further validation in normal cells and animal models.
- These results are based solely on in vitro experiments; further in vivo animal studies are needed to evaluate the efficacy and safety of these extracts.
Response: Thank you for your valuable comment. We fully acknowledge this important limitation and appreciate the reviewer's insight. We are pleased to report that we have recently received funding to initiate a follow-up preclinical study using xenograft mouse models of colorectal cancer to evaluate both the anticancer efficacy and safety profiles of these extracts simultaneously. These investigations will provide crucial data on therapeutic efficacy and potential adverse effects, which will be reported in our subsequent publications as part of the translational pathway toward clinical development.
While our current findings are based on in vitro experiments, previous studies have demonstrated preliminary safety profiles for both extracts in animal models. Syzygium aromaticum ethanolic extract showed no mortality or signs of toxicity in albino rats at doses up to 1000 mg/kg for three weeks, with no significant changes in biochemical or hematological parameters (1). Similarly, S. aromaticum essential oil demonstrated no acute toxicity in Galleria mellonella larvae at concentrations up to 10 mg/ml (2), while DMC from S. nervosum was reported to be non-toxic to ICR mouse models (3). However, we recognize that comprehensive in vivo studies specifically targeting colon cancer efficacy and safety are essential.
In accordance with the reviewer’s suggestion, we have added a statement to the Conclusion section (Page 21-22, lines 884-888,) of the manuscript. Nevertheless, we acknowledge that comprehensive in vivo studies specifically designed to evaluate both the efficacy and safety of these extracts in colon cancer models are crucial for advancing their translational potential.
References:
- Saeed TA, Osman OA, Amin AE, El Badwi SM. Safety assessment and potential anti-inflammatory effect of ethanolic extract of Syzygium aromaticum in albino rats. Advances in Bioscience and Biotechnology. 2017 Nov 16;8(11):411.
- Vasconcelos PG, Abuna GF, Raimundo e Silva JP, Tavares JF, Costa EM, Murata RM. Syzygium aromaticum essential oil and its major constituents: Assessment of activity against Candida spp. and toxicity. Plos one. 2024 Jun 18;19(6):e0305405.
- Utama K, Khamto N, Meepowpan P, Aobchey P, Kantapan J, Sringarm K, Roytrakul S, Sangthong P. Effects of 2′, 4′-Dihydroxy-6′-methoxy-3′, 5′-dimethylchalcone from Syzygium nervosum seeds on antiproliferative, DNA damage, cell cycle arrest, and apoptosis in human cervical cancer cell lines. Molecules. 2022 Feb 9;27(4):1154.
- Use of words such as "upstream railway", "hyphenated" distracts the reader and appears using words not generally used in these types of scientific manuscripts.
Response: We thank the reviewer for pointing out this important stylistic concern. We agree that the use of unconventional expressions such as “upstream railway” and “hyphenated” is inappropriate in the context of a scientific manuscript and may distract readers from the scientific content.
In response, we have carefully revised the manuscript to improve clarity and consistency in tone. Specifically, the phrase “upstream railway” has been replaced with “incidence steadily rising,” and the sentence on Page 2, lines 45–46 has been restructured accordingly. The term “hyphenated” has also been removed or replaced with more appropriate terminology.
Additionally, we conducted a thorough review of the manuscript to ensure that all wording aligns with standard scientific writing conventions.
We thank the reviewer again for this valuable suggestion, which will help shape and strengthen the direction of our future research.

Reviewer 3 Report
Comments and Suggestions for Authors
To the authors:
In the study titled: “Anticancer activities of the ethanolic extract from Syzygium aromaticum and Syzygium nervosum via cell cycle arrest and apoptosis in colorectal cancer cells” T. Yimsoo et Al aims to investigate the potential anti-tumor effects of clove hydrodistillation residue extract from Syzygium aromaticum (SA) and seed extract from Syzygium nervosum (SN) in colorectal cancer cells. Specifically, the study investigates their antioxidant properties and ability to induce cell cycle arrest as possible mechanisms underlying their anticancer activity. However, the current version of the manuscript demonstrates several limitations as the experiments lack sufficient rigor and reproducibility, and need more robust experimental validation.
Major points for revision include:
1. The manuscript requires a text revision, particularly in the results section. The title should be revised. The data description is overly detailed with methodological specifics, which is more appropriate for the Materials and Methods section. The Results should instead focus on clearly presenting and interpreting the experimental findings. Moreover, the description of the antioxidant activity results (2.2.2 paragraph) in the manuscript currently looks like a direct repetition of the data presented in the table, lacking clear interpretation and flow. It is recommended that this section be revised to provide a graphical summary that highlights the comparative efficacy of the SA and SN extracts, emphasizing the biological relevance of the differences in IC50 values. This will improve clarity and help readers better understand the significance of the findings. There is also a discrepancy between figure number and the referred text in line 106 “while S. nervosum was shown in Figure 2”.
Furthermore, the table needs to be reformatted.
Based on the aforementioned test limitation authors should consider ad just the text making the manuscript more fluent and coherent with the reported results.
2. In section 2.2.2. the authors discuss about the antioxidant activity of SA and SN extracts by DPPH and ABTS assay. It could be more appropriate to show the quantified data indicating % of antioxidant capacity by reporting Trolox equivalent antioxidant capacity (TEAC, mM) as indicated in the M&M section. Moreover, the authors should represent the IC50 value with a graph showing the IC50 value versus different compounds.
⸻
3. The author should compare the effects of these extracts with the efficacy of standard chemotherapy treatments in colorectal cancer cells, both as standalone treatments and in potential combination. This would help evaluate the feasibility of a combinatorial therapy aimed at reducing the required dose of chemotherapy for colorectal cancer patients. Additionally, the study should assess and compare the therapeutic efficacy of the natural compound-based treatment versus standard chemotherapy. Furthermore, these experiments should also be conducted on healthy cells to evaluate any potential side effects induced by the tested compounds.
4 In section 2.4.1, the authors demonstrate that the treatment induces early apoptosis in a dose-dependent manner. They should modify the gating strategy to obtain appropriate quantification. Moreover, there is no correspondence between the quantification and the flow cytometry plot in Figure 3C. Furthermore, the authors need to improve the quality of both the bar plot and dot plots to better visualize the differences between the treatment and the axis label, respectively.
5 The quality of the microscopy images shown in Figures 4A and 4B is suboptimal although the caspase activity appears to be detectable upon treatment. There is a high amount of background fluorescence, which makes it difficult to appreciate the specific signal. Authors should improve image quality and include higher-magnification images. Furthermore, authors should perform, the quantification of fluorescence intensity with greater methodological rigor.
6. In this work, the authors suggest that this compound may induce cell cycle arrest and apoptosis, suggesting the involvement of mechanisms leading to cell cycle blockade. Have the authors considered performing additional immunostaining analyses for DNA damage markers or real-time PCR for selected genes related to cell cycle arrest and DNA damage?
7. The regions labeled as sub-G1 likely correspond to cellular debris rather than apoptotic events characterized by DNA fragmentation. To properly assess cell cycle phase distribution analysis authors should provide representative dot plots in addition to histogram overlays. This allows a more precise assessment of the gating strategy and the population included in the analysis. Furthermore, it is strongly recommended that the authors use a dedicated cell cycle analysis algorithm—such as those based on the Watson or Dean-Jett-Fox models
8. The quality of the Western blot is unsatisfactory. The bands are poorly defined, with significant background and nonspecific signals, and seem not to be coherent with the reported quantification.
Comments on the Quality of English LanguageThe English is generally good, but several sections need reorganization for improved clarity
Author Response
Dear Reviewer,
Thank you for the opportunity to review this manuscript. I sincerely appreciate the trust you have placed in me.
The submitted manuscript, entitled “Anticancer Activities of the Ethanolic Extracts from Syzygium aromaticum and Syzygium nervosum via Cell Cycle Arrest and Apoptosis in Colorectal Cancer Cells” by Thunyatorn Yimsoo et al., has been submitted for consideration as an article in the International Journal of Molecular Sciences. In accordance with your recommendation, we have carefully addressed the reviewers’ comments and made corresponding revisions to the manuscript.
Below are my major comments and suggestions for further revision. Kindly consider the following points:
- The manuscript requires a text revision, particularly in the results section. The title should be revised. The data description is overly detailed with methodological specifics, which is more appropriate for the Materials and Methods section. The Results should instead focus on clearly presenting and interpreting the experimental findings. Moreover, the description of the antioxidant activity results (2.2.2 paragraph) in the manuscript currently looks like a direct repetition of the data presented in the table, lacking clear interpretation and flow. It is recommended that this section be revised to provide a graphical summary that highlights the comparative efficacy of the SA and SN extracts, emphasizing the biological relevance of the differences in IC50 values. This will improve clarity and help readers better understand the significance of the findings. There is also a discrepancy between figure number and the referred text in line 106 “while S. nervosum was shown in Figure 2”.
Furthermore, the table needs to be reformatted.
Based on the aforementioned test limitation authors should consider adjust the text making the manuscript more fluent and coherent with the reported results.
Response: We sincerely appreciate the reviewer’s thoughtful comments and constructive suggestions regarding the clarity and structure of the antioxidant activity section. In Section 2.2.2, we have revised the text to avoid redundancy with the data table and to improve the narrative flow by integrating interpretative commentary. We now highlight the comparative efficacy of Syzygium aromaticum (SA) and Syzygium nervosum (SN) extracts relative to the positive control (vitamin C), focusing on the biological relevance of differences in ICâ‚…â‚€ values. To enhance clarity and facilitate data interpretation, we have included a graphical summary (Figure 2, Page 8) in the form of a bar chart comparing the ICâ‚…â‚€ values of each extract and the standard. Additionally, the discrepancy between the in-text citation and figure number (line 106) has been corrected (Page 3, lines 107), and all figure references throughout the manuscript have been reviewed for consistency. These revisions were made to ensure that the manuscript presents a more fluent and coherent account of the results.
- In section 2.2.2. the authors discuss about the antioxidant activity of SA and SN extracts by DPPH and ABTS assay. It could be more appropriate to show the quantified data indicating % of antioxidant capacity by reporting Trolox equivalent antioxidant capacity (TEAC, mM) as indicated in the M&M section. Moreover, the authors should represent the IC50 value with a graph showing the IC50 value versus different compounds.
Response: We sincerely thank the reviewer for the valuable suggestion. We apologize for the inconsistency in the original manuscript. Although the Materials and Methods section initially stated that a Trolox standard curve was used to quantify antioxidant capacity, we would like to clarify that vitamin C was employed as the reference standard in both the DPPH and ABTS assays throughout this study. The manuscript has been revised accordingly to reflect this correction (Page 20, lines 796–797). As a result, the antioxidant capacity is not expressed as Trolox equivalent antioxidant capacity (TEAC, mM), but rather assessed through ICâ‚…â‚€ values using vitamin C as the positive control. This choice was based on the availability and consistent response of vitamin C under our experimental conditions. In response to the reviewer’s recommendation, we have also included a bar graph (Figure 2) representing the ICâ‚…â‚€ values of Syzygium aromaticum (SA), Syzygium nervosum (SN), and vitamin C. This graphical representation facilitates a clearer comparison of antioxidant efficacy among the tested compounds.
- The author should compare the effects of these extracts with the efficacy of standard chemotherapy treatments in colorectal cancer cells, both as standalone treatments and in potential combination. This would help evaluate the feasibility of a combinatorial therapy aimed at reducing the required dose of chemotherapy for colorectal cancer patients. Additionally, the study should assess and compare the therapeutic efficacy of the natural compound-based treatment versus standard chemotherapy. Furthermore, these experiments should also be conducted on healthy cells to evaluate any potential side effects induced by the tested compounds.
Response: We sincerely thank the reviewer for this insightful comment. We acknowledge the importance of comparing the anticancer efficacy of Syzygium aromaticum (SA) and Syzygium nervosum (SN) extracts with standard chemotherapeutic agents in colorectal cancer to assess their potential for reducing chemotherapy dosage while maintaining therapeutic efficacy. In the present study, our primary objective was to investigate and compare the anticancer efficacy of Syzygium aromaticum (SA) and Syzygium nervosum (SN) extracts in colorectal cancer cell lines. The findings demonstrated that SA exhibited superior efficacy compared to SN. However, we did not include standard chemotherapeutic agents in this initial phase, as the focus was on elucidating the cellular mechanisms of action of both extracts. We greatly appreciate the reviewer’s recommendation regarding the inclusion of a standard chemotherapy drug for comparative purposes. In response, we are pleased to report that we have recently received funding to conduct in the next phases. This will include:
- In vitro experiments using normal cells to assess selective cytotoxicity,
- In vivo experiments using animal models to evaluate therapeutic efficacy and systemic safety,
- And the use of 5-fluorouracil (5-FU) as a positive control for direct comparison with the plant extracts.
These upcoming studies will allow us to systematically compare the therapeutic efficacy and safety profiles between the extracts and standard chemotherapy. Furthermore, comprehensive toxicity assessments including both acute and chronic toxicity studies can clearly define the potential side effects and safety margins of the tested compounds. This data will be crucial for evaluating the feasibility of these extracts as alternative or adjunctive therapies in colorectal cancer treatment in the future.
- In section 2.4.1, the authors demonstrate that the treatment induces early apoptosis in a dose-dependent manner. They should modify the gating strategy to obtain appropriate quantification. Moreover, there is no correspondence between the quantification and the flow cytometry plot in Figure 3C. Furthermore, the authors need to improve the quality of both the bar plot and dot plots to better visualize the differences between the treatment and the axis label, respectively.
Response: We sincerely thank the reviewer for highlighting this important issue. In response, we have addressed each of the points raised as follows:
- Gating strategy: We have revised the gating approach to improve the accuracy of apoptotic population quantification. We consistently used the control sample to define our gates in each experiment, ensuring reliable and reproducible results. Before gating, we performed fluorescence compensation for both dyes using single-stained controls, which allowed us to accurately correct for spectral overlap between the two fluorochromes. The cut-off values distinguishing control (negative) and positive populations (using the highest extract concentration) were carefully set and adjusted based on these controls before gating, and we applied these settings consistently throughout our analysis.
- Consistency between plots and quantification: We carefully reviewed the data and confirmed that the updated dot plots and corresponding bar graphs are now aligned. The quantification values presented in the revised figure match the representative dot plots, resolving the previous discrepancy.
- Figure improvements: The quality of both the dot plots and bar plots has been enhanced. Specifically, we have:
- Increased image resolution and clarity;
- Standardized axis labels and quadrant markers;
- Included properly scaled error bars and statistical significance annotations.
These improvements are reflected in the revised Figure 4C (Page 10) and detailed in the Results section (Page 9, lines 272–291). We believe these revisions enhance data transparency and facilitate better interpretation of the apoptotic responses.
- The quality of the microscopy images shown in Figures 4A and 4B is suboptimal although the caspase activity appears to be detectable upon treatment. There is a high amount of background fluorescence, which makes it difficult to appreciate the specific signal. Authors should improve image quality and include higher-magnification images. Furthermore, authors should perform, the quantification of fluorescence intensity with greater methodological rigor.
Response: We sincerely thank the reviewer for the constructive comments regarding the microscopy image quality and fluorescence quantification. In response, we have addressed each issue as follows:
- Image clarity and background fluorescence: We have enhanced the quality of Figures 5A and 5B by carefully adjusting image contrast and reducing background intensity without altering the original fluorescence signal. These modifications improve signal visualization while preserving data integrity. During analysis, fluorescence quantification was based on a predefined threshold using the negative control group to eliminate background interference.
- Image magnification: We acknowledge the suggestion regarding higher magnification. In the current version, we maintained a 50 µm scale bar to visualize caspase activation across a representative cell population, enabling more robust and reproducible quantification. This magnification is consistent with prior reports in IJMS and related journals.
- https://www.mdpi.com/1422-0067/24/7/6082
- https://www.mdpi.com/1422-0067/25/1/49
- https://www.mdpi.com/1422-0067/22/9/4433
- https://pmc.ncbi.nlm.nih.gov/articles/PMC11357354/
- Fluorescence quantification: Fluorescence intensity was measured using ZEN 3.5 (Blue Edition) software from multiple random fields across three independent experiments. The results are now reported as mean fluorescence intensity with appropriate statistical analysis (including standard error and significance testing), thereby ensuring greater methodological rigor and reproducibility.
These improvements are reflected in the revised Figure 5 (Page 11) and the updated Results section.
- In this work, the authors suggest that this compound may induce cell cycle arrest and apoptosis, suggesting the involvement of mechanisms leading to cell cycle blockade. Have the authors considered performing additional immunostaining analyses for DNA damage markers or real-time PCR for selected genes related to cell cycle arrest and DNA damage?
Response: We appreciate the reviewer’s insightful suggestion regarding the inclusion of additional mechanistic analyses such as immunostaining for DNA damage markers or real-time PCR of genes related to cell cycle arrest. We fully agree that further investigation into the underlying molecular mechanisms, particularly DNA damage response pathways and gene regulation, is crucial for a more comprehensive understanding of the observed anticancer effects. However, such mechanistic studies were beyond the scope and timeline of the current experimental phase due to project and resource limitations.
At present, we have received funding support to conduct follow-up in vivo studies. As part of this next phase, we plan to perform immunostaining for DNA damage using the TUNEL assay on tumor tissue sections obtained from treated animals. This approach was intentionally designed because oral administration of the extract in animal models allows the incorporation of pharmacokinetic processes that are absent in in vitro settings, thereby providing a more physiologically relevant context. The findings from this ongoing study will be reported in a subsequent article.
- The regions labeled as sub-G1 likely correspond to cellular debris rather than apoptotic events characterized by DNA fragmentation. To properly assess cell cycle phase distribution analysis authors should provide representative dot plots in addition to histogram overlays. This allows a more precise assessment of the gating strategy and the population included in the analysis. Furthermore, it is strongly recommended that the authors use a dedicated cell cycle analysis algorithm—such as those based on the Watson or Dean-Jett-Fox models
Response: We sincerely thank the reviewer for the valuable feedback regarding our cell cycle analysis and the interpretation of the sub-G1 population. We address the reviewer’s concerns as follows:
- Gating and debris exclusion: In our initial gating strategy, debris and dead cells were excluded by plotting FSC-A vs SSC-A to isolate the main cell population. Additional doublet discrimination was performed using FSC-A vs FSC-H plots to ensure that only single, viable cells were included in the analysis. These gating steps were consistently applied across all samples.
- Sub-G1 and apoptosis correlation: Although debris was excluded for standard cell cycle analysis, we observed a marked increase in sub-G1 events following treatment, especially at higher extract concentrations. This sub-G1 increase paralleled the results from our Annexin V/PI staining, which confirmed apoptosis. Based on this correlation and widely accepted practices, we interpreted the sub-G1 population as indicative of apoptotic DNA fragmentation rather than nonspecific debris.
- Quantification strategy and model fitting: We attempted to apply both the Watson and Dean-Jett-Fox models in FlowJo for automated fitting. However, neither model provided satisfactory fits across all experimental conditions. Consequently, we used the control group showing well-defined G1, S, and G2/M peaks to manually define the gates, which were then uniformly applied across all treatments. This manual gating approach is widely used when model fitting fails to achieve accurate phase separation.
- Supporting literature: The use of sub-G1 as a marker of apoptosis is consistent with several studies, including those published in IJMS and other peer-reviewed journals:
- https://www.mdpi.com/1422-0067/14/8/16970
- https://www.mdpi.com/1422-0067/16/12/26089
- https://www.mdpi.com/1422-0067/14/1/850
- https://pubmed.ncbi.nlm.nih.gov/28966729/
We believe that our combined approach excluding debris via gating, confirming apoptosis via Annexin V/PI staining, and referencing literature provides a robust and reliable interpretation of the sub-G1 population as reflective of treatment-induced apoptotic cell death. We addressed the revision in Figure 7 (Page 13-14).
- The quality of the Western blot is unsatisfactory. The bands are poorly defined, with significant background and nonspecific signals, and seem not to be coherent with the reported quantification.
Response: We sincerely thank the reviewer for pointing out the concern regarding the quality of the Western blot images. In response, we have taken the following corrective measures:
- Repeat and enhancement of the blot: We repeated the Western blot experiment for p21 and enhanced the image quality by adjusting contrast and sharpness to improve band definition. These adjustments were carefully applied to preserve the integrity of the original data and improve alignment between the blot and the quantitative analysis.
- Image–quantification consistency: The updated image more clearly reflects the band intensities reported in the bar graph, thereby improving the visual correspondence between qualitative and quantitative data. These revisions are reflected in the updated manuscript (Page 13, lines 464–473) and the revised Figure 8 (Page 14).
- Residual technical limitations: While the overall quality has improved, minor dot-like artifacts remain within the p21 bands, likely due to incomplete blocking. However, these do not interfere with the interpretation of the results or alter the quantification. Such technical issues have been previously documented in the literature and may result from antibody concentration, blocking conditions, or washing steps [1,2].
References
- Mahmood, T.; Yang, P. C. Western Blot: Technique, Theory, and Trouble Shooting. Am. J. Med. Sci. 2012, 4, 429–434.
- Bass, J. J.; Wilkinson, D. J.; Rankin, D.; Phillips, B. E.; Szewczyk, N. J.; Smith, K.; Atherton, P. J. An Overview of Technical Considerations for Western Blotting. Biol. Methods 2017, 4, e72.
We thank the reviewer again for this valuable suggestion, which will help shape and strengthen the direction of our future research.

Round 2
Reviewer 2 Report
Comments and Suggestions for Authors
None.
Author Response
Thank you very much for your comment.
Reviewer 3 Report
Comments and Suggestions for Authors
To the authors:
Although the manuscript effectively demonstrates the potential of this study, particularly highlighting the anticancer role of Syzygium aromaticum (SA) and Syzygium nervosum (SN) in p53 wild-type colorectal cancer cells compared to p53 mutated cancer cells, unfortunately, the revised version of this manuscript failed to add the requested robustness and improve the requested sections. In fact, although the authors answered most of the question trough literature support, they missed to add new experiment that should increase the robustness and significance of each data. Unfortunatly, mainly due to budget limitations, most of the request from the first revision remains unanswered. Here I’m reporting all the limitation I still found in the revised version of the manuscript and between the experiment of the revised paper, for example in the revised manuscript, the statistical sample size appears limited, with only one cell line per category. While the potential of these natural compound extracts is reported, the additional experiments requested to strengthen the robustness and significance of the results are missing from the revised version, primarily due to budget and material constraints. In the manuscript, as mentioned in the revision request, including the same treatment in a healthy cell line—albeit one different from colon healthy cell lines—would further reinforce the findings. This is due to the challenges of maintaining colon-healthy cell lines in culture and the complexities of the media required. Additionally, although the main goal of this work is to demonstrate the effects of these natural compound-derived extracts, exploring the potential for combinatorial treatment with standard chemotherapy could enhance the translational value of this research for future strategy development. Unfortunately, these experiments are also lacking due to budget limitations.
Some questions raised during the first revision remain unanswered, and the results are not fully satisfactory or clear, addressing these issues is important for publication in this journal. A list of unsatisfactory questions is reported below:
- A supplementary figure has been added to highlight the potential antioxidant role of Syzygium aromaticum (SA) and Syzygium nervosum (SN) in colorectal cancer (CRC) cell lines, which enhances the interpretation of the data. However, the integration of this content in Section 2.2 still needs to be made more fluent.
- All abbreviations should be defined in full length upon first use to ensure clarity for the reader (e.g., "LC-DAD-ESI-MS/MS" should be introduced as "liquid chromatography coupled to diode array detection and electrospray ionization tandem mass spectrometry").
- The tables remain poorly formatted and should be revised to improve readability and consistency.
- I understand the authors focus on elucidating the cellular mechanisms of Syzygium aromaticum (SA) and Syzygium nervosum (SN) in colorectal cancer (CRC) cell lines. However, the absence of comparisons with standard chemotherapeutic agents used in colorectal cancer remains a limitation in evaluating the clinical relevance and translational potential of the findings. If the inclusion of chemotherapeutic controls is not feasible at this stage, testing the extracts on additional CRC cell lines and healthy non-cancerous cells would still significantly strengthen the robustness of the data. This would help define the selectivity and specificity of cytotoxicity, thus providing a clearer indication of the therapeutic window, making the manuscript's message more translational for future applications.
- I appreciated the clarifications on the gating strategy used in the flow cytometry analysis. However, it remains essential to include an additional representative dot plot showing the apoptotic profile after treatment with SA at 200 µg/mL. Despite the mention of fluorescence compensation, spectral overlap still appears to affect data interpretation. A new, fully compensated dot plot at this key concentration would help validate the gating and support accurate quantification of apoptosis.
- The authors state that molecular analyses (e.g., gene expression or DNA damage markers) were not performed and that in vivo studies are planned. While this is understandable due to resource and timeline limitations, the lack of such data represents a notable gap, especially since the manuscript proposes cell cycle arrest and apoptosis as the main mechanisms of action.
- The requested representative dot plots for cell cycle analysis were not provided, and the suggested use of analysis algorithms (e.g., Watson or Dean-Jett-Fox models) was not implemented. This limits the ability to verify the accuracy of gating and quantification. In addition, the sub-G1 peak in the histograms appears very close to the Y-axis baseline, making it difficult to distinguish the apoptotic population from background noise or debris. Improved visualization and modeling, including the representation of the gating strategy used to select alive cells, are essential to enhance the reliability of the cell cycle data. I acknowledge that the accumulation of a sub-G0/G1 peak is related to apoptosis, but I would like clarification on the accuracy of the cell cycle analysis since the distribution of cells does not allow for the algorithm's application.
In conclusion, the manuscript is coherent and demonstrates significant potential, especially concerning the anticancer effects of SA and SN, as well as their biologically relevant impacts, which are clearly detailed. However, the response to the first revision is still somewhat unsatisfactory. There was an opportunity to clarify certain aspects and include mechanisms related to robustness or, further, the translational significance of the findings. Unfortunately, these elements were constrained by budgetary and material limitations.
Comments on the Quality of English LanguageQuality of english is generally good
Author Response
Dear Reviewer,
Thank you for the opportunity to review this manuscript. I sincerely appreciate the trust you have placed in me.
The submitted manuscript, entitled “Anticancer Activities of the Ethanolic Extracts from Syzygium aromaticum and Syzygium nervosum via Cell Cycle Arrest and Apoptosis in Colorectal Cancer Cells” by Thunyatorn Yimsoo et al., has been submitted for consideration as an article in the International Journal of Molecular Sciences.
In accordance with your recommendation, we have carefully addressed the reviewers’ comments and made corresponding revisions to the manuscript.
Below are my major comments and suggestions for further revision. Kindly consider the following points:
1. A supplementary figure has been added to highlight the potential antioxidant role of Syzygium aromaticum(SA) and Syzygium nervosum (SN) in colorectal cancer (CRC) cell lines, which enhances the interpretation of the data. However, the integration of this content in Section 2.2 still needs to be made more fluent.
Response: We sincerely thank the reviewer for this valuable comment. In response, we have revised Section 2.2 to enhance the narrative flow by integrating the antioxidant-related findings more coherently. This revision provides a more logical connection between the figure 2 and the main results, thereby improving the overall clarity and interpretation of the findings (Page 7, Lines 196–208).
2. All abbreviations should be defined in full length upon first use to ensure clarity for the reader (e.g., "LC-DAD-ESI-MS/MS" should be introduced as "liquid chromatography coupled to diode array detection and electrospray ionization tandem mass spectrometry").
Response: We thank the reviewer for pointing this out. In response, the abbreviation “LC-DAD-ESI-MS/MS” has now been defined in full as “liquid chromatography coupled to diode array detection and electrospray ionization tandem mass spectrometry” upon its first appearance in the Results section (Page 3, Lines 106–107).
3. The tables remain poorly formatted and should be revised to improve readability and consistency.
Response: We sincerely thank the reviewer for this valuable comment. In response, we have thoroughly reformatted all tables throughout the manuscript to improve readability and ensure consistency. Adjustments include standardizing column alignment, font size, row spacing, and footnote formatting. We also refined the table layouts to enhance visual clarity and facilitate comparison between groups. The updated tables are included in the revised manuscript.
4. I understand the authors focus on elucidating the cellular mechanisms of Syzygium aromaticum (SA) and Syzygium nervosum (SN) in colorectal cancer (CRC) cell lines. However, the absence of comparisons with standard chemotherapeutic agents used in colorectal cancer remains a limitation in evaluating the clinical relevance and translational potential of the findings. If the inclusion of chemotherapeutic controls is not feasible at this stage, testing the extracts on additional CRC cell lines and healthy non-cancerous cells would still significantly strengthen the robustness of the data. This would help define the selectivity and specificity of cytotoxicity, thus providing a clearer indication of the therapeutic window, making the manuscript's message more translational for future applications.
Response: We sincerely thank the reviewer for this valuable suggestion, which highlights a key aspect regarding the translational relevance and therapeutic selectivity of our findings. In response, we have conducted additional cytotoxicity testing using MRC-5 cells, a non-cancerous human lung fibroblast line commonly employed as a model for normal proliferating cells. These data are now presented in the Results section (Page 7–8, Lines 227–247) and further discussed in the Discussion (Page 16, Lines 619–636).
The inclusion of MRC-5 data allowed us to calculate the Selectivity Index (SI), which is now presented in Figure 3 and Table 3. These values demonstrate that both SA and SN extracts exhibit preferential cytotoxicity toward colorectal cancer cells, particularly HCT116, over normal cells. This strengthens the robustness of our findings and provides an initial indication of a favorable therapeutic window.
Although the direct inclusion of standard chemotherapeutic agents (e.g., 5-FU) was not feasible due to resource limitations, we fully acknowledge this as an important comparative benchmark. Accordingly, we have addressed this limitation in the revised Conclusion (Page 22, Lines 915–932) and propose it as a key direction for future studies.
5. I appreciated the clarifications on the gating strategy used in the flow cytometry analysis. However, it remains essential to include an additional representative dot plot showing the apoptotic profile after treatment with SA at 200 µg/mL. Despite the mention of fluorescence compensation, spectral overlap still appears to affect data interpretation. A new, fully compensated dot plot at this key concentration would help validate the gating and support accurate quantification of apoptosis.
Response: We sincerely thank the reviewer for this valuable and constructive comment, and we apologize for the oversight in our previous submission. As rightly pointed out, the dot plot for the SA 200 µg/mL treatment lacked proper fluorescence compensation, which may have affected the interpretation of the apoptotic profile.
In response, we have now reanalyzed the data and included a newly generated, fully compensated representative dot plot of HT-29 cells treated with SA extract at 200 µg/mL (Page 9, Lines 289–308). (Figure 4C). This analysis was performed using single-stained controls to ensure appropriate fluorescence compensation and eliminate spectral overlap between Annexin V–FITC and PI. The updated dot plot now clearly delineates the viable, early apoptotic, late apoptotic, and necrotic cell populations, thereby strengthening the accuracy and reliability of our flow cytometry data.
6. The authors state that molecular analyses (e.g., gene expression or DNA damage markers) were not performed and that in vivo studies are planned. While this is understandable due to resource and timeline limitations, the lack of such data represents a notable gap, especially since the manuscript proposes cell cycle arrest and apoptosis as the main mechanisms of action.
Response: We greatly appreciate the reviewer’s thoughtful comment highlighting the importance of molecular analyses in supporting the proposed mechanisms of action. We fully acknowledge that gene expression profiling and DNA damage markers would provide deeper mechanistic insights, particularly regarding cell cycle arrest and apoptosis.
Although such analyses could not be included in the current study due to time and resource constraints, we have performed Western blotting to assess the expression of key cell cycle regulatory proteins, p21 and cyclin D1, in response to SA and SN treatment. These findings provide supportive evidence for the observed cell cycle arrest effects, though we recognize they do not fully delineate the underlying molecular pathways.
To avoid overinterpretation of the data, we have revised the manuscript title, key word, and discussion to clarify that the observed effects are associated with, rather than mechanistically confirmed to involve, cell cycle arrest and apoptosis. Specifically, the manuscript title has been revised from:
“Anticancer Activities of the Ethanolic Extracts from Syzygium aromaticum and Syzygium nervosum via Cell Cycle Arrest and Apoptosis in Colorectal Cancer Cells”
to:
“The Effect of the Ethanolic Extracts from Syzygium aromaticum and Syzygium nervosum on Antiproliferative Activity and Apoptosis in HCT116 and HT-29 Cells”
We believe this revision more accurately reflects the scope of our data and aligns the conclusions with the available experimental evidence.
7. The requested representative dot plots for cell cycle analysis were not provided, and the suggested use of analysis algorithms (e.g., Watson or Dean-Jett-Fox models) was not implemented. This limits the ability to verify the accuracy of gating and quantification. In addition, the sub-G1 peak in the histograms appears very close to the Y-axis baseline, making it difficult to distinguish the apoptotic population from background noise or debris. Improved visualization and modeling, including the representation of the gating strategy used to select alive cells, are essential to enhance the reliability of the cell cycle data. I acknowledge that the accumulation of a sub-G0/G1 peak is related to apoptosis, but I would like clarification on the accuracy of the cell cycle analysis since the distribution of cells does not allow for the algorithm's application.
Response: We thank the reviewer for the detailed and constructive comments. In response, we have now included representative plots for cell cycle analysis, with clear delineation of the sub-G1, G0/G1, S, and G2/M phases. Additionally, the flow cytometric gating strategy is described as follows:
Flow cytometric gating strategy for cell cycle analysis
Representative plots showing the sequential gating strategy applied to analyze cell cycle distribution. The gating steps were performed as follows:
1. Singlet discrimination: Cells were gated on FSC-H vs. FSC-A plots to exclude doublets and select singlet populations, ensuring accurate quantification of cell cycle phases. Forward and side scatter plots were further used to isolate the main cell population based on size and granularity.
2. Exclusion of technical artifacts: PE-W vs. PE-A plots were employed to eliminate events with abnormal width-to-area ratios, indicative of technical artifacts. Only well-defined single-cell events were included for subsequent analysis as presented by scatter plots.
3. Cell cycle phase determination: Histograms were generated from the gated population (P3) to assess DNA content. Manual gating was performed to identify the sub-G1, G0/G1, S, and G2/M populations, despite challenges in peak separation.
4. Quantification summary:
The percentage of events within each gate relative to total events, confirming successful gating and representative distribution.
5. Baseline comparison: Untreated control cells were used as the reference for defining normal cell cycle distribution. The increase in the sub-G1 population observed with increasing concentrations of the extract indicates DNA-fragmented cells, as confirmed by apoptotic cells stained with Annexin V-FITC and PI with flow cytometry (Figure 4 in manuscript). Furthermore, the results of the cell cycle and apoptosis assays (flow cytometry) were consistent. Therefore, we believe that this figure is adequate to demonstrate that our compounds induce an increase in cell death during the cell cycle process, which is confirmed by apoptosis. In order to reduce misunderstandings in the communication and reporting of the results, we have replaced the term "apoptotic cells" with "DNA-fragmented cells" when reporting the cell cycle results (Sections Result, Page 13, Lines 478-479). In the discussion, we also include a discussion of this issue (Sections Discussion, Page 17, Lines 699-703).
Regarding the use of analysis algorithms: we acknowledge the suggestion to apply models such as Watson or Dean-Jett-Fox; however, due to irregularities in DNA content distribution, particularly the overlapping of peaks and the proximity of the sub-G1 population to the Y-axis baseline, algorithmic modeling was not applicable in this dataset.
We hope that the inclusion of representative plots and clarification of our gating strategy addresses the reviewer’s concerns regarding the reliability and interpretability of the cell cycle data.
We thank the reviewer again for this valuable suggestion, which will help shape and strengthen the direction of our future research.

Round 3
Reviewer 3 Report
Comments and Suggestions for Authors
The authors have considered the comments and integrate the requests received during the review process. They have provided additional information enhancing the clarity of manuscript and overall, the quality of the work. In particular, the authors have updated figures highlighting and clarifying the potential antioxidant role of Syzygium aromaticum (SA) and Syzygium nervosum (SN) in colorectal cancer (CRC) cell lines.
The authors have now included a fully compensated representative dot plot showing the apoptotic profile after treatment. While the requested representative dot plots for cell cycle analysis and the application of analysis algorithms (e.g., Watson or Dean-Jett-Fox models) were not implemented due to technical limitations, the authors have improved the visualization of the sub-G1 peak and included a detailed gating strategy to select live cells, thereby enhancing the reliability of the cell cycle data.
Regarding the clinical relevance and translational potential of the findings, the authors recognize the limitation resulting from the lack of comparison with standard chemotherapeutic agents used in CRC treatment. Due to resource and time limitations, the inclusion of such controls was not feasible at this stage. However, to increase the robustness of the data, additional experiments were performed on healthy non-cancerous cells, MRC5. This analysis highlights the differences observed between healthy cells and CRC cell lines, although no difference is noted between the MRC5 cell line and the p53 wild-type HT-29 cancer cell line. The results should specifically focus on the differences observed between the p53-mutated CRC cell line and the healthy non-cancerous cell line. They should integrate a quantitative plot representing the SA and SN effects at a representative working concentration across the different cell lines. Emphasizing the differential response between healthy cells and p53-mutated cancer cell lines provides more relevant insight into the selectivity and potential therapeutic targeting, which is critical for the translational impact of the study.
Author Response
1. The results should specifically focus on the differences observed between the p53-mutated CRC cell line and the healthy non-cancerous cell line. They should integrate a quantitative plot representing the SA and SN effects at a representative working concentration across the different cell lines. Emphasizing the differential response between healthy cells and p53-mutated cancer cell lines provides more relevant insight into the selectivity and potential therapeutic targeting, which is critical for the translational impact of the study.
Response: Thank you for this valuable suggestion. To address the reviewer’s comment regarding the importance of highlighting differences between p53-mutant colorectal cancer (CRC) cells and non-cancerous cells, we have incorporated a new quantitative bar graph in Figure 4 (Page 9), along with corresponding revisions in the Results (Page 8, Lines 243–253) and Discussion (Page 16-17, Lines 657–685) sections.
The updated figure compares the ICâ‚…â‚€ values and Selectivity Index (SI) at 72 hours for both Syzygium aromaticum (SA) and Syzygium nervosum (SN) across three cell lines: HCT116 (p53 wild-type), HT-29 (p53 mutant), and MRC-5 (normal fibroblast).
